# ALLaM: Large Language Models for Arabic and English

**M Saiful Bari**[*] **Yazeed Alnumay**[*]
**Norah A. Alzahrani    Nouf M. Alotaibi    Hisham A. Alyahya**
**Sultan AlRashed    Faisal A. Mirza    Shaykhah Z. Alsubaie**
**Hassan A. Alahmed    Ghadah Alabduljabbar    Raghad Alkhathran**
**Yousef Almushayqih    Raneem Alnajim    Salman Alsubaihi**
**Maryam Al Mansour Majed Alrubaian    Ali Alammari    Zaki Alawami**
**Abdulmohsen Al-Thubaity    Ahmed Abdelali    Jeril Kuriakose**
**Abdalghani Abujabal**[*] **Nora Al-Twairesh**[*] **Areeb Alowisheq**[*] **Haidar Khan**[*]

National Center for AI (NCAI)
Saudi Data and AI Authority (SDAIA)
Riyadh, Saudi Arabia

## Abstract

We present ALLaM: **A**rabic **L**arge **La**nguage **M**odel, a series of large language models to support the ecosystem of Arabic Language Technologies (ALT). ALLaM is carefully trained considering the values of *language alignment* and *knowledge transfer* at *scale*. Our autoregressive decoder-only architecture models demonstrate how second-language acquisition via vocabulary expansion and pretraining on a mixture of Arabic and English text can steer a model towards a new language (Arabic) without any catastrophic forgetting in the original language (English). Furthermore, we highlight the effectiveness of using parallel/translated data to aid the process of knowledge alignment between languages. Finally, we show that extensive alignment with human preferences can significantly enhance the performance of a language model compared to models of a larger scale with lower quality alignment. ALLaM achieves state-of-the-art performance in various Arabic benchmarks, including MMLU Arabic, ACVA, and Arabic Exams. Our aligned models improve both in Arabic and English from their base aligned models. Arabic assets are released in Hugging Face. [1]

## 1 Introduction

Language modeling has significantly progressed from its humble origins, transitioning from fundamental probabilistic methods to complex neural priors. The foundational work by Shannon (1951) on the information theory of language laid the groundwork for predicting the next word in a sequence, which was subsequently tackled by Bengio et al. (2003) in neural networks. The field experienced a substantial leap with the introduction of LSTMs (Hochreiter & Schmidhuber, 1997) in language models (LM) (Peters et al., 2018b), which could capture longer dependencies in LMs but proved difficult to scale. The emergence of scalable and distributed architectures like Transformers (Vaswani et al., 2017) and the potential for precisely (Kaplan et al., 2020; Hoffmann et al., 2022) compressing web-scale data has resonated in recent years with the advancements of *Generative Pretraining* (Radford et al., 2018; Brown et al., 2020a; Anil et al., 2023).

With the release of ChatGPT (OpenAI, 2022), followed by the introduction of more frontier class models Gemini (Google, 2024), Claude (Anthropic, 2022), Reka (Ormazabal et al., 2024), Mistral (Mistral, 2024), Llama-3 (Meta, 2024) and recently released Qwen-2 (Yang et al., 2024), large language models have demonstrated significant leaps over each generation of models (Laskar et al., 2023). This exponential growth in performance has raised hope in the possibility of achieving Artificial General Intelligence (Hendrycks & Mazeika, 2022; Marcus, 2022). This rapid advancement has spurred discussions across various fields, including ethics, economics, and technol-

---

[*]Core contributors
[1]https://huggingface.co/ALLaM-AI

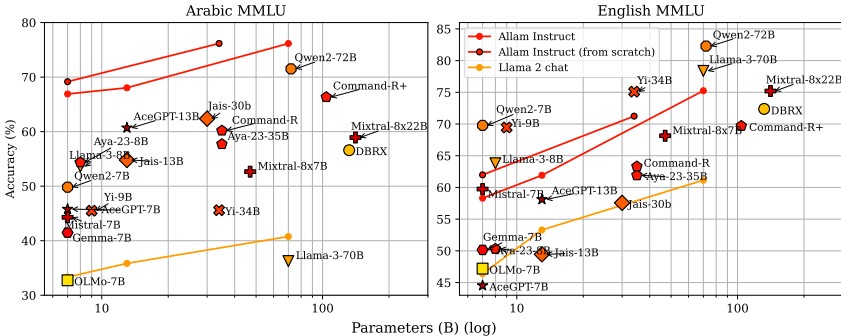

Figure 1: Performance on Arabic (Koto et al., 2024) and English (Hendrycks et al., 2020) MMLU Benchmarks. `ALLaM` (red line) shows impressive improvement from its base model `Llama-2` (yellow line). All evaluations were done on the latest version of the fine-tuned (chat or instruct) models. The `ALLaM` 7B from scratch model also shows significant improvement over the `ALLaM` 7B continued pretraining model.

ogy (Weidinger et al., 2021). Judging from the initial capabilities (Bubeck et al., 2023), the potential of these frontier models are reinventing the way humans interact with machines, impacting social norms, productivity, trends, and culture on a broader scale (Zhou et al., 2024). However, most of these frontier-class models are primarily trained on English and often lack a connection to localized regional cultures and norms (Naous et al., 2024). This gap has the potential to result in slow and irreversible manipulation of regional identities and lead to cultural homogenization.

The natural course to reverse this trend is to invest resources in curating data and building models to support the diversity of languages and cultures represented in the modern world. While this is possible, the significant training costs of LLMs and their environmental impact have become major concerns in recent years (Strubell et al., 2019). The vast computational resources required to train LLMs contribute to substantial carbon emissions (Luccioni & Hernandez-Garcia, 2023). Governments [2] and non/for-profit organizations (Dodge et al., 2022; Google, 2021; Amazon, 2021), are increasingly aware of these issues. This awareness has led to discussions about the ethical implications of AI development and the need for sustainable practices concerning "*When and how to scale the training of these models*". In addition, curating data for each language/region at pretraining scale is also a difficult task, since most available data comes from a few high-resource languages.

To address these concerns, we consider the problem of adapting strong, but potentially under-trained, open pretrained models, rather than starting from a randomly initialized model. Technically, this involves continuing training of a model in a new language to facilitate Second Language Acquisition (SLA) (Swain & Lapkin, 1995), popularized by Bari et al. (2020) in NLP and recently adapted to LLMs by Nguyen et al. (2023). This process involves the challenging task of incorporating an additional language distribution without compromising the source language(s). For instance, if a pretrained model was initially trained in English, expanding to an additional language presents challenges related to tokenization. Figure 2 gives an overview of `ALLaM`'s tokenizers compared to a tokenizer primarily trained on English.

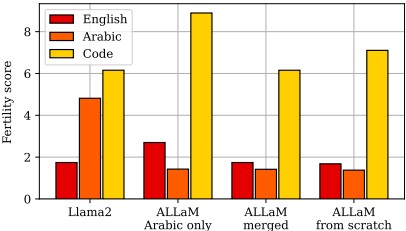

Figure 2: Comparison of tokenizer fertility scores. The chart illustrates the fertility scores across four tokenizers: `Llama-2`, `ALLaM` Arabic only, `ALLaM` merged with `Llama-2`, and `ALLaM` Arabic/English (from scratch model). We calculate the fertility over a random subsample of the entire English, Arabic, and code training corpus.

There are various Arabic LLMs that have been developed, such as `Jais` (Sengupta et al., 2023) and `AceGPT` (Huang et al., 2023), which trained from scratch or continued training `Llama-2`, respectively. A more detailed description and other relevant works can be found in Appendix C.

Our approach to building `ALLaM`, large language models developed specifically for fluency and understanding in Arabic and English, can be outlined as follows. We first demonstrate the feasibility

---

[2] https://www.cnrs.fr/en/update/jean-zay-supercomputer-recycling-its-heat

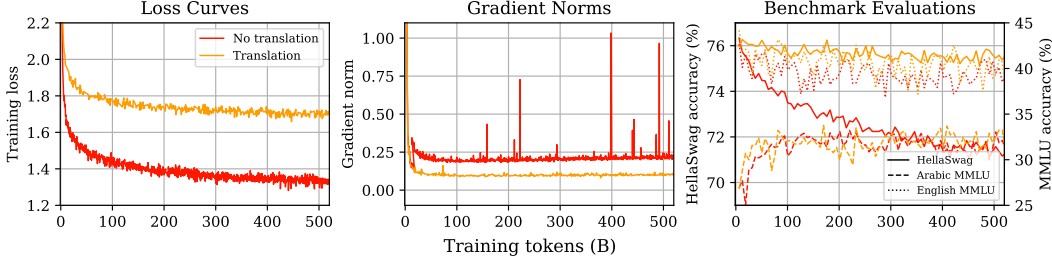

Figure 3: Measuring the effect of adding machine translated Arabic data to pretraining. Although the two loss curves look normal (*left*), adding the translated Arabic reduced the frequency of gradient spikes during training (*center*). Adding translated Arabic data also clearly helps align the Arabic and English capabilities of the model and reduce catastrophic forgetting (*right*).

of adapting an existing pretrained English model (`Llama-2` (Touvron et al., 2023a)) to fluency in both Arabic and English through *tokenizer and vocabulary expansion*. Then, we apply our learnings to train a stronger model from scratch (random initialization) in a similar fashion, i.e., pretraining on English followed by training in mixed Arabic and English. The resulting model exhibits impressive performance and has favorable tokenization properties compared to other models. This approach aligns with both our technical goals and our commitment to sustainable practices. Our overall contributions are summarized below:

1. The `ALLaM` model series, with the goal of supporting the cultural values of the Arabic-speaking world. We train four models at four different scales: 7B, 13B, 34B and 70B models initialized by `Llama-2` weights along with 7B and 34B models trained from scratch.

2. Our model achieves state-of-the-art results in Arabic, as well as improving overall English performance of the original `Llama-2` model. Refer to Figure 1 for an overview.

3. We demonstrate that it is possible to train highly-performant models in low-resource languages from publicly available model weights using our continued pretraining recipe with tokenizer expansion, presenting a path for better representation of low-resource languages.

4. The training methodology and decision-making involved in training the LLM. We provide necessary ablation studies for most crucial decisions.

## 2 PRETRAINING

Pretraining language models on trillions of natural language tokens represents the bulk of the cost required to build an effective language model. This large investment of time and compute precludes experimentation or ablation for every decision. Thus, before starting to train `ALLaM` from random initialization, or "scratch", we experiment in the continue-pretraining regime. As the name implies, continue-pretraining is the practice of warm-starting an experiment from an already pretrained LM.

We begin by discussing our entire pretraining corpus, describe experiments conducted with continue-pretraining, and finally describe pretraining from scratch.

### 2.1 PRETRAINING DATA

For English, many high quality and large scale datasets are available for pretraining (Together Computer, 2023; Soldaini et al., 2024; Gao et al., 2021; Penedo et al., 2023). We harnessed subsets from RedPajama (Together Computer, 2023), FineWeb (Penedo et al., 2024), Dolma-v1 (Soldaini et al., 2024) and Pile (Gao et al., 2021) datasets e.g., Dolma-CC, The Stack (Kocetkov et al., 2022), PeS2o, PubMed, DM-Math (Saxton et al., 2019) and StackExchange (Soboleva et al., 2023). In total, we had access to 4T high to medium quality English tokens for pretraining. For our 30B pre-training we sample 5.2T tokens from the RedPajamaV2 (Together Computer, 2023). We provide detailed description of our data filtering method in the Appendix G.

Pretraining data in the Arabic language is much more limited, thus we undertook large scale collection and curation of Arabic language data. This includes in-house crawled sources covering Web documents, news articles, books (literature, religion, law and culture, among others), Wikipedia (over 1M articles), and audio transcripts (books and news). To ensure high quality Web data, we applied the following processing steps: *(i)* drop documents with language identification score below 95%, *(ii)* drop short documents that are less than 30 words, *(iii)* drop documents with duplicate

URLs or high ratio of spam and stop words, and *(iv)* drop duplicate documents using exact matching. We experimented with fuzzy matching but opted against using it, as it was too restrictive given the scarcity of Arabic data. During Arabic data processing, no standardizing or normalization of characters was performed. For language identification, we used fastText embeddings. For spam and stop words selection, we used Mubarak et al. (2020). The majority of the Arabic data is from the web, which naturally contains Arabic varieties; however, no specific Dialectal Arabic (DA) datasets were intentionally included in the corpus. Knowledge of the distribution of Arabic varieties would require the data to be classified into DA, and since, to the best of our knowledge, DA identification is a hard task that is still under research. According to Abdul-Mageed et al. (2024); Bouamor et al. (2019); Abdelali et al. (2021), classifying the data to DA is difficult. Nonetheless, for future work, we plan to evaluate ALLaM's ability to understand and generate DA via ALDi (Keleg et al., 2023). For Audio transcription, we used the SauTech ASR system.

Additionally, we extended our Arabic data with translated English content using an in-house machine translation system. We translated the following English datasets from Dolma: Wikipedia, books, C4 and peS2o, which also are part of our English data. The hypothesis is that this will improve Arabic-English language alignment, leading to a better Arabic model. Figure 3 demonstrates the impact of Arabic translated datasets in the pretraining data mixture [3]. While models trained without translated data exhibit lower training loss, those trained with translated data show more stable training, as evidenced by fewer spikes in gradient norms. Incorporating Arabic translated data in the pretraining dataset mitigates catastrophic forgetting in English. In total, we curate 540B Arabic tokens[4] of which 270B are natural Arabic tokens and 270B are translated Arabic tokens.

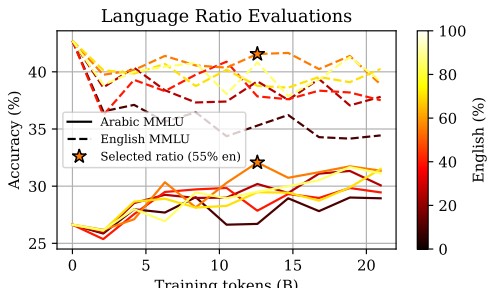

Figure 4: We determine the optimal Arabic/English language mixture that balances between acquiring Arabic understanding while retaining English proficiency by conducting ablations over 6 Arabic/English ratios (trained up to 20B tokens). We found that a 45/55 Arabic/English ratio achieves the best performance, as measured by English and translated Arabic MMLU.

For Arabic, we have a total of ∼540B tokens, and for English, we have a total of ∼660B distinct tokens. Based on the sampling ratio from Table 1, we collected the training data from the corpus.

**Data Mixture** To build a performant model in both Arabic and English, we conducted experiments to determine the optimal language mix. Figure 4 gives an overview of data-mixture experiments on our curated Arabic-English corpus. We conducted the experiments with the same sampling ratio (Table 1) and data order. We observe best trend in performance with 45/55 Arabic/English data mix.

Table 1 shows the language and category mixing distributions for English, Arabic natural, Arabic translated and final mix.

Table 1: ALLaM's pretraining data mixtures. The first four columns summarize the distribution of the continued pretraining mixed Arabic/English data. The English only pretraining from scratch mixture is shown in the last column. We upsample data to match the mixture rates when needed. Mixed English is a subset of English only.

| Domain | Mixed Arabic & English | | | | English Only |
|---|---|---|---|---|---|
| | English | Arabic | | Mixed | |
| | | Natural | Translated | | |
| Web | 31% | 71% | 65% | 48% | 71% |
| Books | 9% | 13% | 12% | 11% | 3% |
| Wiki | — | 0.70% | 0.61% | 0.3% | 0.1% |
| News | — | 14% | — | 3% | — |
| Science | 16% | — | 22% | 14% | 6% |
| Code | 39% | — | — | 21% | 17% |
| Math | 5% | — | — | 2.5% | 0.9% |
| Other | — | 1.3% | 0.39% | 0.2% | 2% |
| Lang Mix | 55% | 22.5% | 22.5% | 100% | 100% |
| Tokens | 660B | 270B | 270B | 1.2T | 4T |

lated and final mix. Following mainstream work (Touvron et al., 2023b; Chowdhery et al., 2022; Rae et al., 2021), web data constitutes the highest ratio with 71%, 65% and 48% of the Arabic nat-

---

[3] Our in-house translation system's BLEU score is around 25.23 on IWSLT test set.
[4] Tokens counted by our merged tokenizer.

ural, Arabic translated and overall mix, respectively. We limited the contribution of English web data to 31%, as the `Llama-2` base model was trained on a significant amount of English web data already, and we expected that increasing its ratio might degrade performance. We ensured that high quality sources, such as books, news, code etc. are well represented in our mixture.

## 2.2 CONTINUED PRETRAINING

Open source and open weight models present an attractive option to conduct pretraining experiments cheaply. However, they also present challenges, since most such models do not natively support Arabic or other languages. We develop a simple approach to enhance any language model with capabilities in new languages (i.e., language expansion). The approach relies on two steps: *(i)* tokenizer augmentation, and *(ii)* expanded vocabulary learning. We demonstrate that this approach leads to minimal degradation of capabilities in the original language.

**Tokenizer Augmentation**   Existing open weight language models (e.g., `Llama-2`) tokenize Arabic (and other languages) poorly, often splitting words down to the character level or even relying on byte-fallback mechanisms for tokenization. This results in: *(i)* inefficient training, as the pretraining corpus size is inflated, *(ii)* unoptimized inference, since the model must generate more tokens per word, and *(iii)* the effective context length is reduced, because it is based on a fixed number of tokens. To address these issues, we use a corpus of text in the target language to train a tokenizer specialized in that language. We then merge the original tokenizer with the language-specific tokenizer. Merging is accomplished by adding all tokens from the language-specific tokenizer that do not exist in the original tokenizer. As shown in Figure 2, this effectively reduces the fertility score in the target language of the merged tokenizer to the level of the language-specific tokenizer. We expanded Llama-2's $32,000$ token vocabulary to $61,568$ tokens.

**Expanded Vocabulary Learning**   Newly added tokens in the merged tokenizer have no associated embedding representations in the pretrained language model's weights. To learn these representations, we experiment with two approaches: *(i)* random initialization and *(ii)* initialization from combined representations of tokens in the original tokenizer. Approach *(ii)* is accomplished by tokenizing each token in the vocabulary of the new tokenizer using the original tokenizer. The associated embedding representations of this tokenization are then averaged and assigned as the vector representation of the new token. Since we work with tokenizers with byte-fallback, such a tokenization is guaranteed to exist. Figure 5 provides an overview of our initialization method. Initializing the new embeddings from the combination of previously learned embeddings gives a significant boost to the learning of a new language.

**Experiment Details**   Starting from `Llama-2` pretrained model weights, we continue pretraining the `ALLaM`-7B and `ALLaM`-13B models on 1.2T tokens, covering both English and Arabic languages. For the `ALLaM`-70B model, we only train up to 600B tokens (using the same data mixture). In all of our continued pretraining experiments, we used the final learning rate of the pretrained language model (usually $3 \times 10^{-5}$). We experimented with approaches to gradually increase and decay the learning rate with limited success, as such models typically exhibited catastrophic forgetting, indicated by significant drops in performance in the original language. We also considered optimizer state warm up, as open-weight models typically do not include the optimizer states, but found this had little effect on performance. Figure 6 provides an overview of adding dropout during continued pretraining. We observe that adding dropout helps the Arabic language, as it acts as a regularizer for the new distribution. However, `Llama-2` was pretrained on 2T tokens without any dropout, and adding dropout negatively impacts the source language performance. Considering this trade-off, we decided not to add dropout in the continued pretraining stage. For Vocabulary expanded model with LLaMa-2 (7B, 13B, 70B models), unlike recent trends  (AI2, 2024), we did not add any alignment data in this stage of training.

## 2.3 PRETRAINING FROM SCRATCH

Following (Hoffmann et al., 2022; Touvron et al., 2023a), training a high-quality model from scratch requires a substantial amount of tokens. Even when pretraining from random initialization, we find it beneficial to train with a high-resource language for trillions of tokens (English) and then continue training with a mixture of Arabic and English tokens. On small scale experiments (with 1B parameter models) we find that beginning training with two languages can sometimes degrade the performance in English or result in slow learning of both language distributions. From this, we hypothesize that low-resource languages are diluted in the large volume of high-resource language data when pretraining from scratch, even with upsampling and careful tuning.

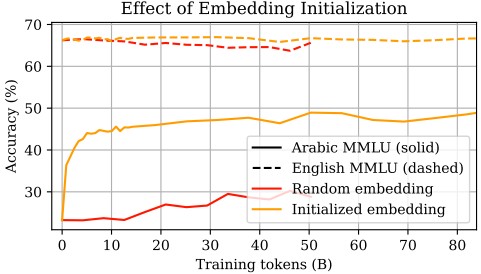
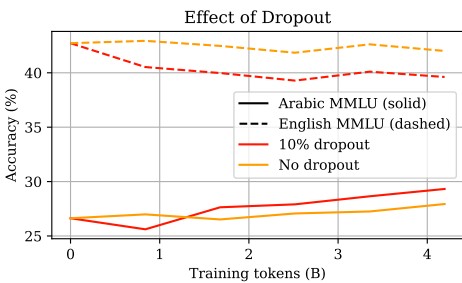

Figure 5: Effect of *random initialization* vs. *embedding initialization* during the start of continued pretraining. We find that initializing the embeddings for new tokens from combinations of existing embeddings speeds up learning dramatically.

Figure 6: Effect of *dropout* during the start of continued pretraining experiments. While introducing dropout can marginally improve the second language acquisition, it negatively impacts the model's capabilities in the original language.

**Training Recipe**   Our pretraining from scratch recipe consists of two steps: training on 4T English tokens followed by training on 1.2T mixed Arabic/English tokens. This retains the English capabilities of the model without catastrophic forgetting, effectively transferring knowledge from one language distribution to another. The only difference between pretraining from scratch and continued pretraining from an existing model is that vocabulary expansion is not required.

We match hyperparameters and architecture for pretraining from scratch with Touvron et al. (2023a), including 4M tokens per batch and max LR $3 \times 10^{-4}$ decayed to $3 \times 10^{-5}$ with a cosine schedule.

During the training of the 7B model (from scratch and vocab expanded models), we found that reducing the learning rate made it difficult to recover without hurting knowledge retrieval/retention capability of the model. To address this, we maintained a constant learning rate during the initial English training phase for the 30B model. In the second stage, we applied a cosine decay learning rate to fine-tune on a mix of English, Arabic, and SFT data. This approach allows for potential extended training with a constant learning rate, followed by shorter but reasonably large-scale cross-lingual alignment using high-quality English, Arabic, and SFT data.

**Data Mixture**   The last column of Table 1 shows the domain mix of ALLaM-7B of the English only pretraining data. For ALLaM-34B, we made slight changes to the domain distribution and improved the quality of the Web data by deploying more aggressive filtering. As expected, web data represent the bulk of the mixture, followed by code and scientific articles.

While our 7B model followed the Arabic-English data mix depicted in Table 1, we made the following changes in ALLaM-34B. We introduced English and Arabic instruction data in our mix (12%). Additionally, we reduced Arabic translated content to 16% from 22.5%. Domain distribution remains close to the one shown in the table. Refer to Appendix H.4 for the training details.

For our 7B model, we adhered to the Arabic-English data mix as outlined in Table 1. However, in ALLaM-34B, we made several adjustments. Specifically, we introduced 12% English and Arabic instruction data into the mix and reduced the proportion of Arabic translated content from 22.5% to 16%. The domain distribution remains largely consistent with what is presented in the table. For further details on the training process, please refer to Appendix H.4.

## 3   ALIGNMENT

Building useful LLMs requires ensuring they are able to follow instructions while adhering to ethical standards and user expectations. This alignment process is especially crucial for models used in diverse linguistic and cultural contexts. In our setting, this means aligning models to the Arabic language and cultural context while also supporting English.

Supervised Fine-Tuning (SFT) (Section 3.1) refines a pretrained model using a carefully selected dataset relevant to specific tasks and domains. Preference training (Section 3.2), on the other hand, aligns the model's outputs with human values and preferences by prioritizing responses that meet user expectations and ethical guidelines. These methods work together to create reliable and ethically sound LLMs for real-world use.

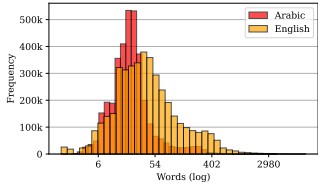 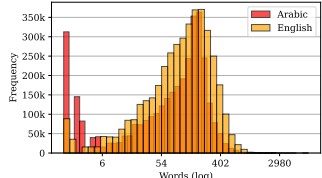 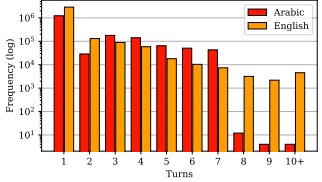

(a) Prompt word count.    (b) Response word count.    (c) Number of conversation turns.

Figure 7: Word count and turn distributions of SFT data. There are two main differences in our Arabic and English SFT datasets: shorter responses are more frequent in our Arabic SFT dataset, while our English SFT dataset contains more dialogues with more than 8 turns.

## 3.1 SUPERVISED FINE-TUNING

**Data** Our SFT data is curated from a diverse array of sources. Given a piece of context from a source, we utilize humans and/or generative models (Ding et al., 2023) to identify if the text can be considered suitable for supervised fine-tuning or if we can generate instructions to create an SFT example from the context. For English, we primarily use public web content as our main source, offering a broad range of high-quality and especially diverse prompts. In contrast, our Arabic data comes from a combination of public and proprietary sources to ensure comprehensive coverage and relevance. To gather data from the source, we collect seed websites or data sources, which involves utilizing domain experts, prompt librarians, local institutes specializing in areas such as Arabic language, history, and politics, the use of commercially permissible licensed LLMs to generate data, and machine translation models to convert rich English SFT data into Arabic. Our datasets cover various domains and capabilities, ensuring the model's proficiency in handling tasks across education, history, Arabic linguistics, politics, religion, computer science, and other fields. The entire Arabic/English collection is called `Ultra-Instinct`, which is not *human generated*, but rather, *human driven*.

**Quality Filtering** Unlike Zhou et al. (2023); AI et al. (2024), we hypothesized that scaling SFT data can unlock diverse capability, as well as improve responsiveness to the prompts. Initially, we crawled the public web for SFT samples. The first version (v1) of `Ultra-Instinct` includes 12M samples

Table 2: Comparison of average word length and lexical diversity of prompts and responses.

| Quality Metric | `Ultra-Instinct` v1 | | `Ultra-Instinct` v2 | |
|---|---|---|---|---|
| | Prompt | Response | Prompt | Response |
| Avg # of Word | 146.94 | 97.19 | 60.81 | 136.47 |
| Lexical diversity | 76.34 | 75.25 | 85.29 | 69.53 |

evenly split between English and Arabic, while the second version (v2), is a reduced version with half the samples. Compared to v1, v2 underwent strict quality checks and human assessments of random subsamples. Our quality checks for v2 included *(i)* assessments based on instruction/response word length, *(ii)* lexical [5] and semantic diversity, exact and near-exact lexical deduplication, *(iii)* removal of low quality machine-translated Arabic data from English sources, and *(iv)* ensuring diversity in questions and commands. For detailed metrics on instruction and response lengths and lexical diversity, see Table 2.

Figures 7a and 7b show the distribution of the prompts and responses in v2, respectively. We focused on maximizing the number of multi-turn conversations in our dataset. Figure 7c shows the distribution of conversation turns from `Ultra-Instinct`.

To extrinsically evaluate the impact of higher quality SFT data, we trained two 13B models using our v1 and v2 SFT datasets. Even though v2 has half as many samples and v1, both versions performed equally well on English and Arabic evaluation benchmarks, as shown in Table 3. This reduction in data volume led to faster

Table 3: Comparative results of `Ultra Instinct` v1 and v2, across various evaluation datasets.

| Version | MMLU | | | Exams (ar) | ACVA | ETEC |
|---|---|---|---|---|---|---|
| | Huang et al. (2023) | Koto et al. (2024) | en | | | |
| `Ultra-Instinct` v1 | 51.0 | 68.0 | 63.8 | 56.8 | 79.8 | 66.8 |
| `Ultra-Instinct` v2 | 51.4 | 68.5 | 63.3 | 56.8 | 76.7 | 65.9 |

training times and reduced costs without compromising performance. It also clearly demonstrates the value of quality filtering for alignment.

---

[5] Lexical diversity is calculated by taking the ratio of the total number of unique words to the total number of words across all samples, excluding stop words.

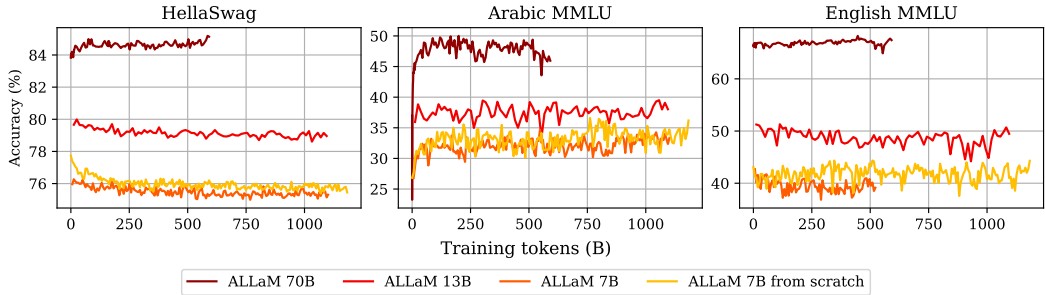

Figure 8: Selected benchmark evaluated through `ALLaM`'s training. Using HellaSwag as a proxy for language understanding, the performance of smaller models degrades when introducing Arabic, while larger models (70B) have enough capacity to improve simultaneously in English and Arabic. Arabic language acquisition is rapid in all models, as indicated by Arabic MMLU.

## 3.2 PREFERENCE TRAINING

After SFT, models are able to converse in multi-turn conversations. However, they are not fully aligned with human preferences. For example, our SFT models were terse and had limited guardrails. To circumvent these issues, we performed preference tuning with human verified samples via Direct Preference Optimization (DPO) (Rafailov et al., 2024).

**Data** The inputs were sourced from early model testers and a manually curated selection of prompts from various domains or attack vectors. These include ethical dilemmas, middle eastern culture, religions, illegal activities, human rights, locale awareness, and personality.

Preference training necessitates both negative and positive outputs for each input. We relied on the testers' feedback to identify the positive outputs. In the absence of positive outputs, we used a model to generate an output and manually verified that the output was aligned. While (Tunstall et al., 2023) utilized preference data from AI Feedback (AIF) at scale, we adopt a more cautious approach in creating preference data. We generate a smaller volume of data, ensuring it is fully reviewed, edited, and verified by humans.

There are two approaches for generating negative outputs: *(i)* on-policy: use the generations of the model we are tuning as negative outputs, and *(ii)* off-policy: use another similar model to generate the negative outputs. We did not verify that the negative outputs were worse than the positive. However, we ensured that the positive outputs were of the highest quality, such that they were almost always better than the negative outputs.

Khan et al. (2023) demonstrated that model outputs can vary significantly depending on the sampling mechanism used. Building on this insight, we generate additional samples for each instance by varying temperature and nucleus sampling techniques. These additional samples are utilized to produce rejected samples, ensuring that `ALLaM` provides more grounded responses and generalizes well across various sampling mechanisms.

In total, we collected 25,854 samples (triplets of {prompt, accepted, rejected}) in English and Arabic language. Using the technique mentioned above, we sample 10 different response from the model to generate additional rejected responses for each sample. This results in a dataset of 245K samples (after filtering) for preference training.

## 4 EVALUATION

In this section, we describe the evaluation of our model and report the results of `ALLaM` 7B, 13B, 34B, and 70B models, as well as other relevant models, such as `GPT-4`, `Command-R+` (Gomez, 2024), and `Jais-30B` (Sengupta et al., 2023). Our evaluations encompass three main types: *(i)* automatic evaluations, *(ii)* LLM-based evaluations, and *(iii)* human evaluations.

**Limitations** Recently, (Alzahrani et al., 2024) showed that multiple choice or cloze test based evaluation may not be robust. In addition, MT-Bench uses an LLM as a judge, and has likely leaked into training datasets. Unfortunately, human evaluation is time-consuming and requires well-trained human evaluators. In this work, we try to ensure robust evaluation and train a balanced assessment of the quantitative metrics and qualitative effectiveness of models in various applications and domains.

Table 4: Arabic benchmark results for instruction tuned models. Follow Table 11 for detailed results.

| | | araSwag | ACVA | MMLU (ar) | | Exams (ar) | ETEC | araTruthfulQA | araMath |
| | | | | Koto et al. (2024) | Huang et al. (2023) | | | | |
| | | 10-shot | 5-shot | 0-shot | 0-shot | 5-shot | 0-shot | 0-shot | 5-shot |
| ALLaM-Instruct | 7B | **49.28** | **80.33** | **66.9** | **49.6** | **52.7** | 62.95 | 36.4 | 36.5 |
| AceGPT-Chat | 7B | 43.4 | 59.35 | 45.8 | 33.58 | 35.57 | 36.05 | **37.9** | 22.5 |
| Llama 2-Chat | 7B | 24.44 | 52.46 | 33.33 | 26.45 | 25.33 | 26.69 | 29.9 | 21.5 |
| Mistral-Instruct-v0.3 | 7B | 30.59 | 60.7 | 44.3 | 34.06 | 31.1 | 34.41 | 30.3 | 26.0 |
| Llama 3-Instruct | 8B | 33.99 | 75.21 | 53.98 | 41.49 | 44.32 | 49.42 | 34.0 | **38.3** |
| ALLaM-Instruct | 13B | 54.77 | **78.59** | **68.11** | **51.03** | **54.93** | 65.59 | **37.5** | **46.8** |
| Llama 2-Chat | 13B | 25.75 | 60.14 | 35.84 | 28.73 | 22.91 | 30.44 | 31.4 | 22.3 |
| Jais-Chat | 13B | **77.12** | 70.68 | 54.8 | 41.43 | 46.93 | 48.68 | 31.6 | 25.3 |
| ALLaM-Instruct | 34B | 59.74 | **81.00** | **75.98** | **60.2** | **58.66** | 74.26 | 35.49 | **46.5** |
| Jais-Chat-v3 | 30B | **88.37** | 70.05 | 62.37 | 30.15 | 51.21 | 38.53 | **37.3** | 32.5 |
| ALLaM-Instruct | 70B | 57.91 | 79.01 | **75.92** | **62.23** | 58.47 | **78.38** | 38.4 | 56.8 |
| Llama 2-Chat | 70B | 30.72 | 59.49 | 40.77 | 32.86 | 28.68 | 30.6 | 32.3 | 25.5 |
| Llama 3-Instruct | 70B | 45.75 | **80.26** | 36.27 | 60.11 | **58.47** | 71.41 | 37.7 | **59.70** |

## 4.1 AUTOMATIC EVALUATIONS

The automatic evaluations cover Arabic and English benchmarks grouped into many categories. Detailed description of the evaluation dataset and benchmark can be found in Appendix I.1.

While serving as a good test bench, observing the dynamics of automatic evaluations during training is also interesting. Figure 8 shows the behavior of selected benchmarks during mixed Arabic/English pretraining while scaling up model size. In particular, we observe that smaller models tradeoff between capability in the new and original languages. However, larger models can simultaneously improve in both languages.

Another observation from automatic evaluations is that some evaluations provide more signal for training decisions than others, e.g., Hellaswag smoothly improves during training while improvements in GSM8k occur in discontinuous jumps. Other benchmarks show no improvement until 1.5T tokens have been seen (i.e., grokking) making them unreliable for early training decisions. Tables 4 and 6 give an overview of the performance of ALLaM instruct models.

In Arabic benchmarks, we can see that ALLaM-70B scores are the best in five (MMLU Arabic (natural and translated), Exams, ETEC, araTruthfulQA) out of the eight benchmark sets. On English, ALLaM is the second-best model in the majority of cases, following Llama 3-Instruct. We highlight the excellent performance of ALLaM on benchmarks released after training was completed (MMLU-Pro, MixEval) and benchmarks the training team did not have access to (ETEC), since they provide a clean evaluation signal. Table 11 and Table 12 in the appendix contain more detailed evaluation results for Arabic and English.

Table 5: MT-Bench scores for Arabic and English. Each score is an average over 80 samples of the score ranging from 0 to 10 returned by the judge (GPT-4).

| Model | English | | | Arabic | | |
| | Avg. | Turn 1 | Turn 2 | Avg. | Turn 1 | Turn 2 |
| AceGPT 13B-chat | 5.44 | 6.76 | 4.12 | 6.33 | 7.01 | 5.64 |
| ALLaM 13B Instruct | 7.34 | 7.67 | 7.01 | 7.57 | 7.9 | 7.23 |
| ALLaM 70B Instruct | 7.44 | **7.91** | 6.96 | **8.19** | **8.4** | **7.97** |
| Jais 13B Chat | 4.18 | 4.39 | 3.96 | 4.72 | 5.07 | 4.36 |
| Jais 30B Chat v1 | 3.89 | 4.13 | 3.64 | 3.54 | 4.13 | 2.95 |
| Jais 30B Chat v3 | 5.86 | 6.25 | 5.47 | 6.28 | 6.78 | 5.78 |
| Cohere Command R+ | 7.41 | 7.63 | 7.18 | 7.97 | 8.28 | 7.65 |
| Cohere Command R | 6.99 | 7.19 | 6.79 | 7.47 | 7.82 | 7.12 |
| DBRX Instruct | 7.16 | 7.33 | 6.98 | 7.83 | 8.19 | 7.46 |
| GPT 3.5 Turbo | **7.55** | 7.79 | **7.31** | 8.12 | 8.39 | 7.84 |

## 4.2 LLM-BASED EVALUATIONS

MT-Bench (Zheng et al., 2024) consists of 80 multi-turn questions to evaluate models' capabilities on complex instruction-following. In addition to the English version, MT-Bench Arabic was created using GPT-4 to translate the original dataset and human annotators to review and align the prompts to Arabic culture. GPT-4 serves as the LLM judge, scoring responses as recommended in (Zheng et al., 2024). Model performance is compared turn by turn, with results shown in Table 5, where ALLaM-70B achieves the best Arabic performance.

## 4.3 HUMAN EVALUATION

We developed an Arabic multi-turn dataset spanning seven domains: linguistics, history, health, politics, coding, entertainment, and ethics, with each domain featuring ten two-turn questions. Human evaluators compared the responses from two models and were asked to choose the winning response with the following instructions:

Table 6: English benchmark results for instruction tuned models. (Follow Table 12 for detailed results.)

| | | AGIEval | MMLU Average | MMLU-Pro | Ethics | TruthfulQA | ARC Challenge | HellaSwag | MixEval Hard | MixEval Standard |
|---|---|---|---|---|---|---|---|---|---|---|
| | | 0-shot | 0-shot | CoT 5-shot | 0-shot | 0-shot | 0-shot | 0-shot | 5/0-shot (base/ft) | 5/0-shot (base/ft) |
| ALLaM-Instruct | 7B | **47.09** | 58.31 | 27.78 | 69.8 | 42.11 | 51.45 | 75.2 | 28.9 | 67.6 |
| AceGPT-Chat | 7B | 26.33 | 44.53 | — | 53.38 | 49.34 | 42.32 | 70.92 | — | — |
| Llama 2-Chat | 7B | 35.55 | 46.4 | 22.87 | 58.88 | 45.32 | 44.28 | 75.52 | 30.8 | 61.7 |
| Mistral-Instruct-v0.3 | 7B | 42.22 | 59.75 | 36.33 | **73.59** | **59.65** | **58.7** | **82.88** | 36.2 | 70.0 |
| Llama 3-Instruct | 8B | 44.35 | **63.82** | **41.32** | 68.07 | 51.72 | 56.83 | 75.81 | **45.6** | **75.0** |
| ALLaM-Instruct | 13B | 48.42 | **61.8** | **34.05** | **76.47** | **57.69** | **55.89** | **81.14** | **37.2** | **72.8** |
| Llama 2-Chat | 13B | 37.73 | 53.3 | 27.19 | 70.52 | 43.95 | 50.17 | 79.66 | — | — |
| Jais-Chat | 13B | 31.45 | 49.46 | — | 64.92 | 39.66 | 46.84 | 77.6 | — | — |
| ALLaM-Instruct | **34B** | **52.47** | **71.24** | **43.61** | **72.84** | **56.27** | **60.15** | **81.25** | — | — |
| Jais-Chat-v3 | 30B | 36.78 | 57.57 | 26.45 | 68.03 | 42.34 | 51.02 | 78.91 | — | — |
| ALLaM-Instruct | 70B | **65.67** | 75.43 | 48.61 | 76.16 | 58.78 | 59.56 | **84.97** | 51.60 | 83.5 |
| Llama 2-Chat | 70B | 46.0 | 61.15 | 35.16 | 68.5 | 52.77 | 54.27 | 82.14 | 38.0 | 74.6 |
| Llama 3-Instruct | 70B | 63.78 | **78.38** | **59.52** | **77.09** | **61.79** | **64.33** | 82.49 | **55.90** | **84.00** |

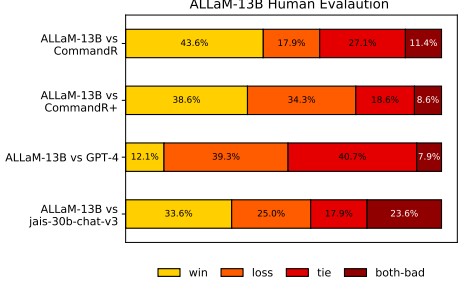

Figure 9: Pairwise win rates as judged by human evaluators. ALLaM-13B wins against many much larger models.

Figure 10: ELO scores from human evaluator preferences. ALLaM is tied with Command-R+ and lags only behind GPT-4.

- Choose a response as the winner if it is the best, tie if both responses are equally good, and both-bad if both responses are not good.

- A response is considered good if it is coherent, grammatically correct, and is a reasonable response to the question or previous turn in the conversation.

- Good responses should be in the correct language (the response should be in the same language as the previous turn, unless another language was requested).

- Good responses should not contain toxicity, hate speech, or bias.

Each pair of responses was inspected by three evaluators, and the winner was determined by majority voting. In case of a tie, a fourth evaluator was used to break the tie. Figure 9 presents the human evaluation results of the pair-wise comparisons of these models: ALLaM-13b, Jais-30b-chat-v3, Command-R-plus, and Command-R-v01. ALLaM-13b's win rate was always higher than its loss rate compared with other models.

Finally, we gather votes from the human evaluators and calculated ELO scores for each model. ELO scoring had two configurations, the default scoring rewards the model for good responses with 1 point, tied responses (good and both-bad) with 0.5 points, and penalizes for bad responses with 0 point. The custom configuration penalizes the model with the bad response and both models if both provided bad responses with 0 point. Figure 10 shows the ELO scores based on the human evaluations. From the figure, GPT-4 achieved the highest score, followed by ALLaM-13b with the second-highest score, outperforming (or matching) larger models such as CommandR+.

## 5 CONCLUSION

The ALLaM model series marks a significant advancement in Arabic Language Technologies, achieving state-of-the-art performance across various Arabic benchmarks while maintaining or enhancing English performance. Through careful training that emphasizes language alignment and transferability, our models demonstrate effective second-language acquisition without catastrophic forgetting. The strategic use of translated data, knowledge encoding, and alignment with human preferences have been crucial in this success.

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

# Contents

# A   ACKNOWLEDGMENT

This release is funded by **Saudi Data & AI Authority**. We thank **Mishari Almishari** and **Yaser Alonaizan** for leading the `ALLaM` project. We list below the different teams that supported the development of ALLaM. **AI Team who joined after the first release of ALLaM models:**

Alhanoof Althnian, Iman Albilali, Jubair Sheikh, Ibrahim Mammu, Ghadah Alsaif, Abdulhameed Alothaimeen, Hadeel Alnegheimish, Emad Alghamdi, Khalid Almubarak, Atheer Al-Barqi, Batool Al-Otaibi, Muhammad Al-Hajji, Abdulraouf Al-Maasoumi, Abdulmajeed Alrowithi, Faisal Qarah, Sakhar Alkhereyf, Arwa Omayrah, Mohammed Al-Saleem, Amal Al-Shammari, Maryam Saif, Raeda Al-Marzoog, Lamyaa Alqhatani, Anas Al-Manie, Ghassan Al-Ward, Shahad Al-Zaidi, Batool Al-Ghoraibi, Alanoud Al-Shammari, Saad Alrazouq.

**Data Acquisition, Management and Engineering:**

Ali Alotaibi, Abdulrahman Alsaudi, Abdulaziz Asheyban, Areej Alokaili, Norah Alangari, Hamad Alnamazi, Fatima Alsalman, Saleh Alrusayyis, Moneerah Alballa, Omar AlGhamdi, Raneem Alqahtani, Amirah Albinhar, Nour Al-Aldahan, Khalid Alharbi, Hanouf Aljlayl, Fatimah Alzubaidi, Lama Aldossary, Rania Alzahrani, Shoug Alkhalaf.

**Development and Infrastructure:**

Saud AlHamoud, Fahad Alothaimeen, Ahmed Alrowatie, Saad Alajlan, Hassan Almitib, Abdinasir Mohamed, Sultan Alotaibi, Mohammed Alanazi , Mohammed Alsharani, Khalid Aljandal, Faisal Abulnaja, Ahmed Alosaimi, Muhannad Abu Hussain, Nasser Almoffarej, Muhammad Al-Hashem, Ahmed Al-Ghamdi, Amer Ashraf, Abeer Al-khars, Nawaf Babgy, Fevicks Kumar, Islam Gamal, Layla Al-Mutairi, Shroq Al-Ghamdi , Amjad Al-Zahrani, Tjad Clark, Ali bjorn, Meshari Alyami, Abdulrahman Bahkali, Reema Alomair.

**Project Management:**

Naif Shalhoub, Esshaq Almotawa, Sara AlRasheed, Mohammed Alshaalan, Mohammed Albreeh, Nezar Kaaki, Muna Alsahli, Abdullah Aldahami, Faisal Al-Tamimi, Tariq Alrouqi, Sahar Alghamdi, Raghad Abuznadah, Naif Almohammed, Oraib Alhemmyaine, Sarah Binlibdah, Shrooq Almohamdi.

**Human Evaluators:**

Amal Almohammadi, Badr almalki, Amjad Alsaeed, Alhanouf Alotaibi, Rajaa Almalki, Maryam Alasmari, Maha Awaji, Fatima Alshehri, Maryam ALshuwaiman, Ebtesam Alzahrani, Yasmeen Algahtani, Atheer Almusallam, Rehab Almalky, Shahad Alsulami, Abdullah Albalawi, Abeer Alqahtani, Lama Alrajeh, Shahad Aqeel, Yasir Alharbi, Rassil Al-Otaibi, Khulud Alawadh, Fatimah Almutairi, Suad Alkhamshi, Abdulrahman Alasmari, Goot Alqahtani, alhanouf Alfoaim, Rawan Aljohani, Aisha Almutairi.

# B   FREQUENTLY ASKED QUESTION

**What is the difference between Alignment at Scale vs. Post-training**   `ALLaM` is trained from a pretrained English language model (LLM). In a pretrained English-only language model, the language alignment between English and Arabic isn't sufficient. By the phrase"... language alignment and transferability of knowledge at scale..." we mean that we focused on English and Arabic co-learning (alignment and knowledge transfer) during the pretraining stage by jointly training both languages together. Note that when we mention "language alignment at scale" here, it refers to the pretraining stage, not the post-pretraining stage (fine-tuning or preference tuning).

**Why did we change the training data distribution for 30B experiments**   Not all the training was done at the same time. As the training progressed, we gained more knowledge about our process, data, and the entire ecosystem of our training engine. Iterating over a single training run incurred significant costs, so we always prioritized quality over ablations for large-scale training runs. Given the available compute and deadline, we were able to conduct only one training run of the 34B model. We discovered that we could apply custom filters to a large data collection based on our use cases and preferences. In the first phase, we used an open-source data collection, and in the second phase,

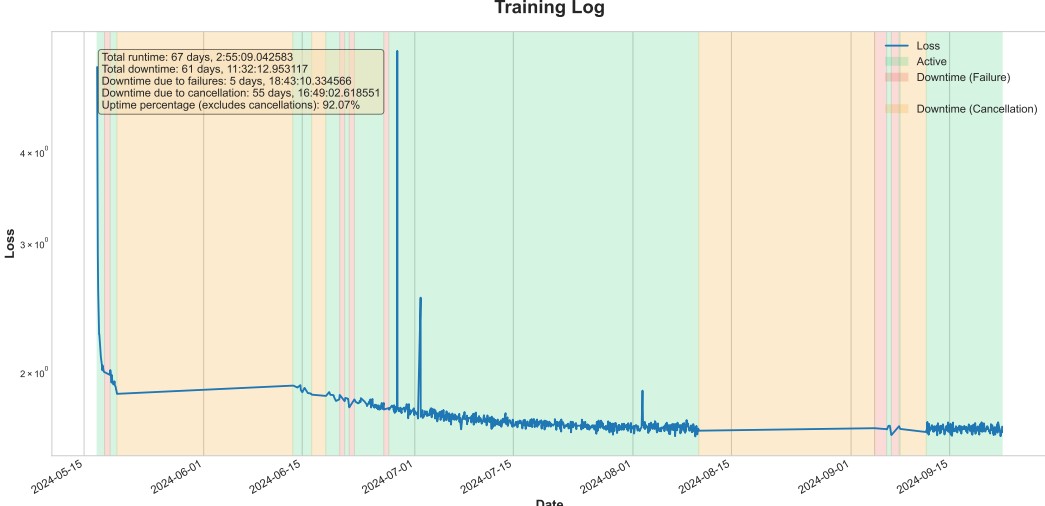

Figure 11: Annotated training log for `ALLaM`-30B showing uptime and downtime. Downtime due to cancellation indicates the training was paused to accommodate other jobs. Failures are marked in red. The model recovered from several loss spikes through the course of training without degrading performance.

we filtered a collection of 84 CC snapshots down to approximately 30T tokens. After filtering, we performed a manual check to verify the data quality.

**What was the uptime of the cluster** We didn't log the uptime of the cluster for all training runs, but we tracked the uptime for our 34B training runs. Figure 11 shows the cluster's uptime.

**Will you opensource/openweight our models?** We released our 13B model on the IBM WatsonX platform in May 2024. Our 7B model, pretrained from scratch, is now available on Microsoft Azure. As time progresses and we build more risk assessment tools with scalable oversight pipeline, we will open up more models to the community.

**Why did we report Instruct model result instead of base model result in the main paper?**

- Blurred distinction between base and instruct models: Modern pretraining often incorporates supervised fine-tuning (SFT) data, including alignment data aimed at improving user interactions. As a result, the clear separation between the base model and the instruct model has become less distinct. Many models today are pre-trained with some degree of alignment, making it difficult to evaluate them purely as base models.

  For instance, as shown in Table 12, the Qwen2-7B-base achieves a score of 77.94 on GSM8k, while ALLaM-7B-base (trained from scratch) achieves 16.98, with a significant delta of 60.96. After supervised fine-tuning, ALLaM-7B-instruct (from scratch) scores 53.6, while Qwen2-7B-instruct scores 77.86, reducing the delta to 24.26. Although AL-LaM shows an improvement of 36.62 points during SFT, Qwen2 experiences no notable gain in performance in this phase. We suspect this is due to the inclusion of alignment data during Qwen2's pretraining phase. Thus, while Qwen2 models are inherently better than ALLaM, the performance gap between Qwen2 and ALLaM at the base level does not necessarily reflect true model capabilities due to the suspected presence of alignment data in the base model.

- Focus on user interaction: The primary goal of building these models is to optimize them for user interaction. Since users will interact with the instruct version of the model, it makes sense to report the results of the model in its instruct phase. This ensures that the reported performance is reflective of the actual experience users will have, making the results more relevant and impactful for the paper's audience.

**What is a fair comparison of different models?**    Many models are released without information on the training dataset size, and of those that do, most do not provide a breakdown of tokens per language. Additionally, it is not clear how we should categorize models, based on *(i)* training FLOPs, *(ii)* model size, *(iii)* inference FLOPs, *(iv)* number of training tokens, or *(v)* quality of training tokens. Therefore, it is extremely difficult to control for all these factors. Instead, we opted to ensure a fair assessment by comparing our 13B models to larger models, such as `Jais` (30B), `Command R+` (104B), and `GPT-4` (unknown, but almost certainly larger than 13B), which actually puts our model at a disadvantage (see Figure 9).

**What are failure cases of `ALLaM`?**    We identified several patterns where the model's performance could be refined through extensive human evaluation. These observations can be summarized as follows:

1. Repetition Issues: In certain versions, the model exhibited a tendency to overgenerate or repeat content. This behavior varied with temperature settings. After extensive testing, we found that a temperature of 0.6 yielded the most balanced outputs.

2. Translation Challenges: The model sometimes regenerates the translation instruction within the translated output, or attempts to summarize the shared context while translating it, rather than focusing solely on the translation itself.

3. Variation in Outputs: When tasked with summarization or translation, the model provides multiple outputs.

4. Precise Instruction Following: The model does not always adhere to specified output length or structure. For instance, when requested to summarize a text in 100 words, it might exceed this limit or produce a summary longer than the input text.

5. Complex Arabic Proofreading: The model faces challenges with certain Arabic-language tasks, such as applying proper punctuation, grammar, and diacritics. While it may provide partially correct answers, it often fails to deliver fully accurate or comprehensive outputs in these areas.

Table 7: `ALLaM` failure examples. For brevity, we omitted parts of the questions/answers with "[...]" to focus on the failure cases

| Question | ALLaM's Answer | Error |
|---|---|---|
| ماهي اختصاصات الأمن الوطني السعودي؟ | اختصاصات الأمن الوطني السعودي تشمل حماية الدولة وصون حدودها [...] وتطوير وتحسين كفاءة البنية التحتية والأداء في مجالات الأمن الوطني والتحريات والمعلومات وتطوير وتحسين كفاءة البنية التحتية والأداء [repeated many times] | Excessive repetition of the same phrase وتطوير وتحسين كفاءة البنية التحتية والأداء through-out the response. We noticed that the KL divergence penalty scalar in DPO impacts this. This issue occurs in fewer than 1% of our total evaluation samples. |
| Have you ever struggled with persuading someone? [...] First, you need to work on your confidence [...] Finally, build trust هلا تساعدني في ترجمتها باللغة العربية [...] | بالطبع! يسعدني مساعدتك في ترجمة المقالة لتتمكن من الاستفادة منها في تدريس طلابك. يرجى تزويدي بنص المقالة وسأعمل على ترجمتها إلى اللغة العربية بكل سرور. | The model asked for context that was already provided. |
| من فضلك، سألقي درس عن آخر آية [...] وأريد ترجمة تفسيرها من تفسير السعدي: الْيَوْمَ أَكْمَلْتُ لَكُرْ دِينَكُرْ بِتمام النصر [...] | في هذا التفسير للآية الكريمة [...] يقول الشيخ السعدي إن هذه الآية تشير إلى أن الدين الإسلامي قد تم تكميله [...] | Instead of translating the provided Tafseer text, the model created a new explanation/summary of it. |
| .فضلا هل يمكنك تقديم خدمة لي بترجمة هذا النص إلى اللغة الإنجليزية؟ تعرف البلاغة على أنها فن الإقناع [...] | Please, can you do me the favor of translating this text into English? Rhetoric is defined as the art of persuasion [...]. | The model translated the request for translation instruction along with the content text. |
| Write a poem in arabic | بالطبع يمكنني مساعدتك في كتابة قصيدة باللغة العربية. إليك قصيدة قصيرة كمثال: في حبّ الوطن يا وطني [...] | ALLaM provided Arabic introduction text instead of starting with English response before the poem. |

**How many Arabic tokens were in `Llama 2`, and how many did you add to `ALLaM`?**    We expanded `Llama 2`'s vocabulary from 32,000 to 61,586 tokens with all the added tokens being Arabic. We define an Arabic token as any token that contains at least one Arabic letter by using the regular expression "`^(?=.*\p{Arabic}).*$`". `Llama 2` contained 46 Arabic Tokens (mostly single Arabic characters), while `ALLaM` contained 29,552 Arabic tokens post-expansion.

Note that Llama 2's tokenizer does not include all Arabic letters, and thus has to rely on byte fallback for less common Arabic letters. For instance, it tokenizes the letter "`<U+0624>`" as two bytes `['<0xD8>', '<0xA4>']`.

## C    RELATED WORK

Our work sits at the cross-section of research building language models that support multiple languages and scaling such techniques in terms of size and data. To successfully train a large language model in a language other than English requires a complete understanding of cross-lingual transferability between languages and a good understanding of scaling laws, as well as the fundamentals of training large language models. In this section, we discuss work on language modeling from the perspectives of cross-lingual alignment, multitask learning, and Arabic specialization.

### C.1    LANGUAGE MODELING AND CROSS-LINGUAL REPRESENTATIONS

In early work, word representations were derived using basic forms of the *skip-gram model* (Mikolov et al., 2013), wherein each word is assigned a representation that does not account for varying contexts (Grave et al., 2018; Pennington et al., 2014). Further work in this area developed word representations that are adaptive to the context surrounding the words (McCann et al., 2017; Peters et al., 2018a; Howard & Ruder, 2018; Devlin et al., 2019; Yang et al., 2019; Radford et al., 2019).

Peters et al. (2018a) introduced ELMo, a model built with a bidirectional LSTM-based language model (LM) for pretraining to obtain contextualized word representations. This technique combines the outputs from all layers linearly when targeting specific tasks. Nonetheless, the sequential nature of LSTM-based LM pretraining presents challenges in scaling training efficiently. Concurrently, Vaswani et al. (2017) developed the *Transformer* architecture, which leverages multi-headed self-attention and positional encoding to handle long-range dependencies and enable parallel processing. Following that Radford et al. (2019) introduced GPT, a model that pretrains a Transformer decoder using a conditional language model objective, with subsequent fine-tuning requiring only minimal modifications. Similarly, Devlin et al. (2019) unveiled BERT, which uses a Transformer encoder pretrained via a masked language modeling (MLM) objective. This approach excelled at task adaptation and benefited from the MLM's ability to encode context bidirectionally, unlike the traditional unidirectional (conditional) LM that processes either the left or right context. Later Raffel et al. (2020) proposed a detailed hybrid encoder-decoder architecture based LLM with an implementation of many objective functions via autoregressive structure.

During the release of BERT, mBERT, a multilingual version of BERT is trained on 102 languages using a shared vocabulary of 110K subword tokens.[6] Despite the lack of explicit cross-lingual supervision, mBERT has demonstrated the ability to learn cross-lingual representations that generalize well across languages. Wu & Dredze (2019); Pires et al. (2019) evaluated the zero-shot cross-lingual transferability of mBERT on several NLP tasks and attributed its generalization capability to shared subword units. Pires et al. (2019) additionally identified structural similarity (e.g., word order) as another crucial factor for successful cross-lingual transfer. K et al. (2020), however, argued that shared subwords contribute minimally, and instead, structural similarity between languages is more critical for effective transfer. Artetxe et al. (2019) further showed that joint training might not be necessary and proposed an alternative method to transfer a monolingual model to a bilingual model by learning only the word embeddings in the target language. They also highlighted the vocabulary size per language as an important factor. Finally, Xue et al. (2021) showed that joint training on a large multilingual vocabulary can robustly map multilingual language models to the same latent space.

In the early days, *Cross-lingual alignment* from mono-lingual embeddings was tricky and often required complex adversarial training (Conneau et al., 2017), careful orthogonal mapping (Artetxe et al., 2018) or semi-supervised learning (Mohiuddin et al., 2020; Bari et al., 2020). With the introduction of mBERT, it became evident that learning joint distribution makes it easier for LLM to achieve cross-lingual alignment at scale. Lample & Conneau (2019) enhanced mBERT by incorporating a conditional LM and a translation LM (leveraging parallel data) objective along with a language embedding layer and trained a larger model utilizing more monolingual data. Huang et al. (2019)

---

[6]`https://github.com/google-research/bert/blob/master/multilingual.md`

suggested employing auxiliary tasks like cross-lingual word recovery and paraphrase detection for pretraining. Subsequent work by Conneau et al. (2019) and Soltan et al. (2022) scaled up the training of multilingual language models. As well, Xue et al. (2021) scale the size and languages in the T5 architecture. In an effort to reproduce GPT-3, Scao et al. (2023) trained the first auto-regressive multilingual LLM.

## C.2 MULTITASK LEARNING AND ALIGNMENT

Early work has demonstrated that multitask learning can enhance the performance of NLP models (Collobert & Weston, 2008). In explicit multitask learning, augmenting all samples during training may introduce noise due to differing output distributions in a traditional full-model fine-tuning setup (Weller et al., 2022; Bari et al., 2021). For implicit multitask learning, Radford et al. (2019) showed that a language model can begin to learn downstream tasks without explicit supervision by pretraining alone. Large language models (Brown et al., 2020b; Smith et al., 2022; Chowdhery et al., 2022) at scale can perform few-shot in-context learning, making them effective multitask models. Additionally, Sanh et al. (2021); Wei et al. (2021); Muennighoff et al. (2022); Chung et al. (2022) found that these implicitly learned language models could be further improved by explicitly fine-tuning them with human instructions and prompts (Bach et al., 2022; Wang et al., 2022) in a multitask fashion. Unlike previous template-based prompting approaches, Ouyang et al. (2022) applied preference tuning with reinforcement learning (Stiennon et al., 2020) using naturally written prompts. Subsequently, Bai et al. (2022) introduced *Constitutional AI* to automate alignment using AI feedback. Recently, following the work of Rafailov et al. (2023), various efforts (Azar et al., 2023; Ethayarajh et al., 2024; Hong et al., 2024; Park et al., 2024; Meng et al., 2024) have been directed towards preference tuning without explicit reward models.

## C.3 LANGUAGE MODELS FOR ARABIC

As of the time of writing, the most prominent Arabic-focused LLMs are:

1. Jais (Sengupta et al., 2023): 13B and 30B base and chat models trained from scratch using a combination of natural and translated Arabic data along with English and code data.
2. AceGPT (Huang et al., 2023): 7B and 13B base and chat models trained from Llama-2 *without* vocabulary expansion.

While Jais and AceGPT are currently the most prominent models, early open models such as AraGPT (Antoun et al., 2020), AraT5 (Elmadany et al., 2022), AraBART (Eddine et al., 2022), and Noon (Lakim et al., 2022) [7] pioneered the area with models developed with limited resources to serve Arabic.

Other models such as Jasmine (Abdul-Mageed et al., 2023) and Aramus (Alghamdi et al., 2023) also showed the need for building a language model for over 400 million speakers worldwide.

In addition to the language adaptation of models and multilingual models reviewed above, recent work has focused on building multilingual/bilingual language models from open weight language models. For example, Ruciński (2024) adapted Mistral 7B for the Polish without vocabulary expansion. Mala-500 is another effort to expand to 534 languages by expanding the vocabulary to 260K tokens and further pretrained Llama-2 using LoRA adaptors (Lin et al., 2024). Due to the number of languages they aimed to support, a small amount of data was included for each language and the evaluation of the approach was limited to measuring perplexity and automatic classification benchmarks. (Cui et al., 2023) introduced a Chinese Language adaptation of Llama and Alpaca models, where the vocabulary was increased to 50K tokens, then continued to pretrain the models and finally fine-tune them.

## D LIMITATIONS

ALLaM was trained on data that may potentially include toxic language, unsafe content, and societal biases originally sourced from the internet, leading to the possible amplification of these biases and

---

[7] https://huggingface.co/Naseej/noon-7b

toxic responses. Although `ALLaM` underwent comprehensive safety training during the alignment phase, more community feedback is needed to iteratively improve `ALLaM`. Additionally, inherent uncertainties in generative models mean that trials cannot encompass every possible use case, making it impossible to predict the model's responses in all contexts. This can occasionally result in inaccurate, biased, or socially unacceptable outputs, even if the prompt itself is not explicitly offensive. Developers must conduct thorough safety evaluations and make specific adjustments to ensure that `ALLaM` is suitable for their intended purposes. Furthermore, the output generated by `ALLaM` should not be considered a statement from `ALLaM`'s creators or any affiliated organization.

## E  ETHICAL STATEMENT

While conducting and presenting this research, we are committed to upholding the highest ethical standards. We recognize the potential impact of large language models on society and the importance of ensuring their responsible development and deployment. Our work adheres to principles of fairness, transparency, and inclusivity, striving to mitigate biases and ensure diverse representation in our training data. We are mindful of privacy concerns and have taken steps to anonymize and secure data used in our research. Additionally, we acknowledge the potential for misuse of language technologies and advocate for their ethical application, promoting beneficial use cases while being vigilant about unintended consequences. `ALLaM` models are made openly available to foster collaboration and further research, with the aim of contributing positively to the advancement of language technologies and supporting the cultural and technological growth of the Arabic-speaking world.

## F  RISK STATEMENT

The deployment and use of LLMs in various applications poses significant risks, including data privacy and security concerns due to the inadvertent inclusion of sensitive information in training datasets. LLMs can perpetuate or amplify biases, resulting in unfair treatment and discrimination in critical decision-making processes. They can also generate convincing but inaccurate content, spreading misinformation and potentially influencing public opinion negatively. Over-reliance on LLMs may diminish human judgment, and the models' susceptibility to adversarial attacks can compromise system integrity. To mitigate these risks, we follow robust governance, continuous monitoring, and iterative improvements. We also adhere to best practices in data handling and model training, fostering transparency and accountability in LLM development.

## G  PROCESSING OF ENGLISH DATA FOR 30B PARAMETER MODEL

The RedPajama V2 [8] (RpV2) dataset is a large resource for training large language models:

**Size and Composition**: RpV2 is an open dataset that includes over 100B text documents sourced from 84 CommonCrawl snapshots. These documents have been processed using the CCNet Wenzek et al. (2019) pipeline, which is known for preparing web-crawled data. The dataset encompasses a massive `30T` tokens, making it, to the best of current knowledge, the largest public dataset released specifically for training language models.

**Quality Signals and Deduplication**: Of the documents included, 30B comes with 40+ precomputed quality signals, and 20B documents are deduplicated. This ensures that the data is not only vast but also of high quality and relevance for training purposes, reducing redundancy, and improving the efficiency of learning.

Please refer to https://www.together.ai/blog/redpajama-data-v2 for the complete list of existing quality signals.

**Multilingual Coverage**: The dataset includes documents in multiple languages, although the primary focus seems to be on English. The other languages are German (DE), French (FR), Spanish (ES), and Italian (IT). Table Appendix G shows detailed statistics of the multilingual breakdown of RpV2 dataset.

---

[8]https://github.com/togethercomputer/RedPajama-Data

Table 8: Document and token counts for different languages

| Language | Document count | Estimated token count (deduped) |
|----------|---------------|-------------------------------|
| English | 14.5B | 20.5T |
| German | 1.9B | 3.0T |
| French | 1.6B | 2.7T |
| French | 1.8B | 2.8T |
| Italian | 0.9B | 1.5T |
| Total | 20.8B | 30.4T |

**Processing Steps Applied**:

Out of the 40+ pre-computed quality signals, we applied the following to create a high-quality subset. Note that we focused on English only.

Table 9: Preprocessing steps to filter high-quality data from the Red-Pajama V2 dataset.

| Annotation Tag | Description | Threshold |
|----------------|-------------|-----------|
| `ccnet_language_score` | Language identification model score. | Keep $\geq 0.6$ |
| `ccnet_length` | Number of characters in the document. | Drop $< 150$ characters |
| `ccnet_nlines` | Number of lines in the document. | Drop $< 3$ lines |
| `rps_doc_ml_palm_score` | FastText classifier prediction for document classification as Wikipedia, OpenWebText, or RedPajama-V1 book (English only). | Sample according to distribution |
| `rps_doc_frac_lines_end_with_ellipsis` | Fraction of lines ending with an ellipsis ("..." or ".."). | Drop $\geq 0.8$ |
| `rps_doc_frac_no_alph_words` | Fraction of words without any alphabetical characters. | Drop $\geq 0.9$ |
| `rps_doc_lorem_ipsum` | Ratio of occurrences of "lorem ipsum" to total characters in content (after normalization). | Drop $\geq 0.5$ |
| `rps_doc_stop_word_fraction` | Ratio of stop words to total words in the document, using stop words from here. | Drop $\geq 0.9$ |
| `rps_doc_symbol_to_word_ratio` | Ratio of symbols ("#", "..." or "..") to words in content. | Drop $\geq 0.9$ |
| `rps_doc_ldnoobw_words` | Count of sequences from the List-of-Dirty-Naughty-Obscene-and-Otherwise-Bad-Words blocklist (see here). | Drop $\geq 0.9$ `(ldnoobw_words/total_words)` |
| `minhash_signature` | Minhash signature for fuzzy deduplication at Jaccard similarity of 0.7, based on 128 hash functions grouped into 14 bands of 9 rows for LSH. | 0.7 `(ldnoobw_words/total_words)` |

**Stats of the filtered data**: Applying all the steps above removed 85.9% of the documents, and 80% of the words.

In addition to the above steps, the CC-NET pipeline, which was used to download CC snapshots, applied language detection and exact-match paragraph deduplication.

Table 10: Document and word counts after each data processing Step

| Step | Input (#docs) | Output (#docs) | Removal Ratio doc level (%) | Input (#words) | Output (#words) | Removal Ratio word level (%) |
|---|---|---|---|---|---|---|
| minhash_signature_0.7 | 24.38B | 24.36B | 0.11 | 2.31T | 2.30T | 0.07 |
| doc_level_exact_duplicates | 24.36B | 14.49B | 40.5 | 2.30T | 1.30T | 43.51 |
| ccnet_language_score | 14.49B | 14.30B | 1.3 | 1.30T | 1.30T | 0.44 |
| ccnet_length | 14.30B | 13.80B | 3.51 | 1.30T | 1.30T | 0.05 |
| ccnet_nlines | 13.80B | 12.87B | 6.79 | 1.30T | 1.28T | 1.37 |
| rps_doc_frac_lines_end_with_ellipsis | 12.87B | 12.87B | 0 | 1.28T | 1.28T | 0 |
| rps_doc_frac_no_alph_words | 12.87B | 12.87B | 0 | 1.28T | 1.28T | 0 |
| rps_doc_lorem_ipsum | 12.87B | 12.87B | 0 | 1.28T | 1.28T | 0 |
| rps_doc_stop_word_fraction | 12.87B | 12.87B | 0 | 1.28T | 1.28T | 0 |
| rps_doc_symbol_to_word_ratio | 12.87B | 12.87B | 0 | 1.28T | 1.28T | 0 |
| rps_doc_ldnoob_w_words | 12.84B | 12.84B | 0 | 1.28T | 1.28T | 0 |
| rps_doc_ml_pal_m_score | 12.84B | 343.12M | 73.29 | 1.28T | 459B | 64.03 |

After applying all processing steps outlined above, we ended up with:

3,431,217,579 (4.3B) total documents, with a total of 4,587,781,981,546 (4.5T) words, and 5.2T tokens.

The average doc length is X words and Y characters.

## H  TRAINING DETAILS

### H.1  COMPUTE AND TRAINING INFRASTRUCTURE

Over the course of our development of ALLaM, we had access to 128-1024 A100 GPUs. Our GPU cluster was equipped with InfiniBand connections to enable high-speed communication between nodes. The all-reduce test on the cluster ranges around 1200-1400 Gbps (node-node interconnect (RoCE)). The entire training period of the models is estimated to be 5M GPU hours.

At the start of the project, we forked Megatron-LM[9] and applied our own customizations (including improving data iterators, adding metadata in the checkpoints, and custom data pipelines). We utilized data, tensor, and pipeline parallelism supported by Megatron-LM to efficiently train at a large scale as well as FlashAttention (Dao et al., 2022; Dao, 2024). By leveraging these techniques, we achieved significant improvements in training speed. The throughput per GPU varied from 135 to 167 TFlop/s/GPU depending on the number of GPUs, number of nodes, batch size, and parallelism strategy. We trained ALLaM with bf16 mixed-precision.

### H.2  PRETRAINING DETAILS

We fine-tune our base model, which was trained on 3.2 trillion (2T Llama-2 + 1.2T ALLaM) tokens, for 3 epochs using Ultra-Instinct-v2 with a learning rate of $5 \times 10^{-6}$ and a batch size of 1024. The model is not trained to generate the prompt, as we mask out our prompt tokens when calculating the loss. Ultra-Instinct-v2 contains a substantial number of multi-turn conversations. To train on these multi-turn conversations, we performed turn-augmentation. Figure 12 visually explains the process of turn augmentation.

While training the SFT model, we encountered tokenization issues. Specifically, Llama-2's tokenizer was trained using sentencepiece[10], which breaks the beginning and end of sequence tokens into multiple tokens and adversely affects long multi-turn conversations. To address this issue, we patched sentencepiece using the HuggingFace LlamaTokenizer wrapper (Wolf et al., 2020). Over many iterations of training, we saw that even having 1% noisy samples (e.g., empty responses or formatting issues) in alignment data can noticeably affect model quality.

---

[9]https://github.com/NVIDIA/Megatron-LM
[10]https://github.com/google/sentencepiece

## H.3 ALIGNMENT TRAINING DETAILS

For DPO, we used a batch size of 512 with $\text{KL}_{penalty} = 0.1$ and a learning rate of $9 \times 10^{-7}$ decayed to $5 \times 10^{-7}$ using a cosine annealing learning rate schedule. We train `ALLaM` for a single epoch using all the preference data.

From our initial experiments with small datasets, we observed issues with model quality even when a small fraction (0.1%) of the data was noisy. In this context, noise can be improper labeling of positive/negative pairs or low quality positive outputs. It is not clear, however, if after scaling up the DPO data whether the model can ignore this type of noise. In early DPO models, trained on data where we did not verify all the samples, we found that even a few moderately noisy samples resulted in broken models that repeatedly generate the same text or output incoherent text.

## H.4 DPO VS. PPO

One of the fundamental differences between DPO and PPO is that PPO is always on-policy with an external reward model. In our experience with DPO, we did not encounter any significant issues with off-policy experiments. Additionally, DPO allows for faster iteration and easier understanding of the training dynamics. The decision to use DPO over PPO was based on logistical constraints rather than a performance comparison of the algorithms. Given our compute setup and time constraints, we chose to proceed with DPO. We plan to explore PPO in the future for alignment.

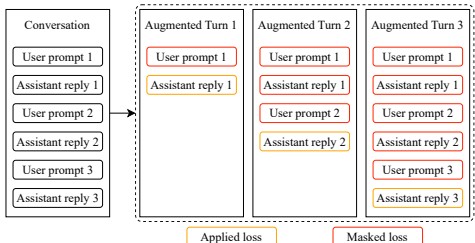

Figure 12: Augmentation process for conversations. The original conversation (left) is expanded into one sample per turn (right), with user prompts and assistant replies marked for training (red) and masking (orange) to enhance `ALLaM`'s language understanding and multi-turn response generation capabilities.

## I EVALUATION

**Evaluation Framework** All evaluations were completed using the Language Model Evaluation Harness (Gao et al., 2023) with the following exceptions: HumanEval was evaluated using BigCode Evaluation Harness (Ben Allal et al., 2022). MMLU-Pro, MixEval, and Arabic MMLU (Koto et al., 2024) were evaluated using the repositories of the dataset creators.

## I.1 DETAILS OF THE EVALUATION DATASETS

The automatic evaluations cover Arabic and English benchmarks grouped into the categories listed below:

1. Multi-domain: MixEval (Ni et al., 2024), MMLU-Pro (Wang et al., 2024), and BBH (Suzgun et al., 2022).

2. Reasoning and Commonsense: HellaSwag (Zellers et al., 2019), PIQA (Bisk et al., 2020), WinoGrande (Sakaguchi et al., 2019), and AraSwag (Nagoudi et al., 2022).

3. World Knowledge and Language Understanding: MMLU (Hendrycks et al., 2020),ARC Easy and Challenge (Clark et al., 2018), TriviaQA (Joshi et al., 2017), BoolQ (Clark et al., 2019), NQ Open (Kwiatkowski et al., 2019), AGIEval (Zhong et al., 2023), Exams-Ar (Hardalov et al., 2020), MMLU Arabic (tr) (Huang et al., 2023), MMLU Arabic (MBZU) (Koto et al., 2024) , and ETEC (in-house curated).

4. Safety and Alignment: Hendrycks Ethics (Hendrycks et al., 2021a), ACVA (Huang et al., 2023), TruthfulQA (Lin et al., 2022), and AraTruthfulQA (in-house curated).

5. Conversation: MT-Bench (Zheng et al., 2024), and Arabic domain capability dataset (in-house curated).

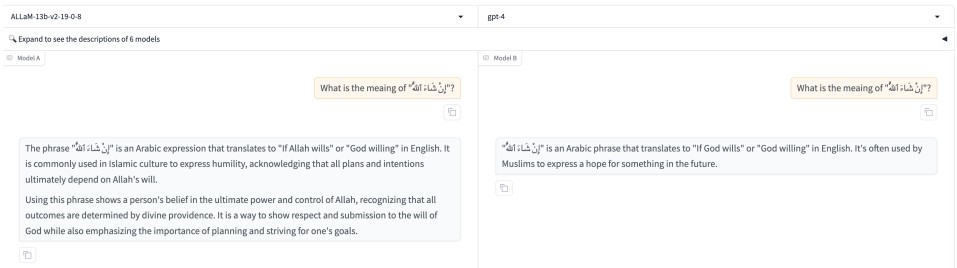

Figure 13: Example of Arabic cultural alignment between `ALLaM` and `GPT-4`.

6. Math: Minerva MATH (Lewkowycz et al., 2022; Hendrycks et al., 2021b), GSM8K (Cobbe et al., 2021) and AraMath (in-house curated).

7. Coding: HumanEval (THUDM, 2022)

The following benchmarks were curated and developed in-house:

- ETEC: a collection of 1891 multiple choice questions covering different exams performed by the Education and Training Evaluation Commission in Saudi Arabia[11].

- AraMath: a set of 600 test samples that were post-processed and prepared from the Ara-Math (Alghamdi et al., 2022) dataset. These samples focus on testing the models' performance on Arabic math problems.

- AraTruthfulQA: a dataset created using similar methodology to the TruthfulQA (Lin et al., 2021) dataset. It comprises a total of 541 samples, 285 samples were translated directly from TruthfulQA using `GPT-4` and carefully validated and localized by human verifiers. Additionally, 256 questions were curated by humans to ensure their contextual relevance and cultural appropriateness.

## I.2 DETAILED ARABIC EVALUATION

Follow the Table 11 for details.

## I.3 DETAILED ENGLISH EVALUATION

Follow the Table 12 for details.

## I.4 EXAMPLES

An example of Arabic culture alignment improvements is illustrated in Figure 13.

---

[11] https://etec.gov.sa/home

Table 11: Comprehensive Arabic benchmark results.

| | | araSwag | ACVA | MMLU (ar) | | Exams (ar) | ETEC | araTruthfulQA | araMath |
| | | | | Koto et al. (2024) | Huang et al. (2023) | | | | |
| | | 10-shot | 5-shot | 0-shot | 0-shot | 5-shot | 0-shot | 0-shot | 5-shot |
| **Pre-trained** | | | | | | | | | |
| **ALLaM-Base (from scratch)** | 7B | 52.68 | 68.46 | 44.45 | 36.28 | 42.09 | 41.7 | 29.4 | 25.5 |
| **ALLaM-Base** | 7B | 51.63 | 66.18 | 41.52 | 34.42 | 38.55 | 36.58 | 29.9 | 11.5 |
| AceGPT | 7B | 46.8 | 59.54 | 36.33 | 27.18 | 32.22 | 25.42 | 30.1 | 19.3 |
| Llama 2 | 7B | 25.62 | 62.93 | 33.61 | 26.64 | 23.09 | 27.85 | 25.7 | 24.8 |
| Mistral-v0.3 | 7B | 30.33 | 53.81 | 40.81 | 32.1 | 31.47 | 32.45 | 27.0 | 16.3 |
| OLMo-1.7 | 7B | 24.44 | 57.8 | 30.97 | 25.7 | 25.7 | 27.17 | 23.5 | 16.8 |
| OLMo | 7B | 22.09 | 56.07 | 31.41 | 24.98 | 28.31 | 23.1 | 26.2 | 31.7 |
| Qwen2 | 7B | 40.26 | 78.74 | 52.91 | 47.16 | 46.0 | 55.23 | 29.9 | 51.2 |
| Gemma | 7B | 25.36 | 54.82 | 46.33 | 26.04 | 22.91 | 25.48 | 24.0 | 39.3 |
| Llama 3 | 8B | 38.95 | 71.54 | 47.62 | 38.88 | 44.69 | 42.86 | 29.9 | 43.8 |
| **ALLaM-Base** | 13B | 54.9 | 77.81 | 51.48 | 40.29 | 47.3 | 44.4 | 28.5 | 17.3 |
| Yi-1.5 | 9B | 28.76 | 61.19 | 46.36 | 34.11 | 34.82 | 40.01 | 24.0 | 44.8 |
| AceGPT-v1.5 | 13B | 48.89 | 73.47 | 42.24 | 33.18 | 40.6 | 33.56 | 30.3 | 18.8 |
| Llama 2 | 13B | 28.63 | 64.52 | 35.83 | 30.0 | 28.86 | 31.13 | 26.2 | 13.8 |
| Jais | 13B | 49.28 | 60.76 | 32.2 | 29.23 | 33.33 | 27.96 | 28.7 | 28.5 |
| **ALLaM-Base** | 70B | **59.35** | **79.67** | 59.21 | 49.34 | 53.82 | 55.97 | 33.5 | 38.7 |
| Jais-v1 | 30B | 54.51 | 68.25 | 37.6 | 32.94 | 43.39 | 34.04 | 29.6 | 19.3 |
| Jais-v3 | 30B | 53.86 | 70.49 | 45.19 | 38.31 | 50.28 | 45.61 | 30.5 | 25.2 |
| Qwen1.5 | 32B | 37.78 | 73.63 | 55.94 | 48.67 | 49.53 | 57.4 | 34.0 | 45.3 |
| Yi-1.5 | 34B | 32.16 | 65.25 | 42.93 | 36.26 | 33.71 | 36.21 | 23.7 | 52.0 |
| Mixtral-8x7B-v0.1 | 47B | 38.43 | 75.64 | 51.25 | 39.74 | 44.32 | 44.61 | 25.5 | 39.8 |
| Llama 2 | 70B | 34.38 | 51.16 | 44.79 | 37.1 | 37.99 | 39.38 | 26.6 | 32.3 |
| Llama 3 | 70B | 54.51 | 74.17 | 36.67 | 59.39 | 55.31 | 64.27 | 31.4 | 53.7 |
| Qwen1.5 | 72B | 44.84 | 76.0 | 61.38 | 54.44 | 54.0 | 62.84 | 34.9 | 51.8 |
| Qwen2 | 72B | 51.76 | 68.7 | **69.94** | **65.0** | 56.98 | 75.16 | **36.0** | 62.3 |
| DBRX | 132B | 47.58 | 72.38 | 53.24 | 47.2 | 47.11 | 51.96 | 26.8 | 49.3 |
| Mixtral-8x22B-v0.1 | 141B | 45.1 | 77.21 | 53.6 | 45.92 | 48.42 | 53.96 | 29.8 | 51.0 |
| **Fine-tuned** | | | | | | | | | |
| **ALLaM-Instruct (from scratch)** | 7B | 50.98 | 79.59 | 69.16 | 51.38 | 52.89 | 67.34 | 30.7 | 42.2 |
| **ALLaM-Instruct** | 7B | 49.28 | 80.33 | 66.9 | 49.6 | 52.7 | 62.95 | 36.4 | 36.5 |
| AceGPT-Chat | 7B | 43.4 | 59.35 | 45.8 | 33.58 | 35.57 | 36.05 | 37.9 | 22.5 |
| Llama 2-Chat | 7B | 24.44 | 52.46 | 33.33 | 26.45 | 25.33 | 26.69 | 29.9 | 21.5 |
| Mistral-Instruct-v0.3 | 7B | 30.59 | 60.7 | 44.3 | 34.06 | 31.1 | 34.41 | 30.3 | 26.0 |
| OLMo-Instruct | 7B | 25.36 | 58.74 | 32.74 | 26.5 | 24.77 | 27.33 | 29.6 | 36.5 |
| Qwen2-Instruct | 7B | 37.78 | 79.3 | 49.82 | 48.07 | 47.3 | 56.18 | 35.1 | 51.3 |
| Gemma-it | 7B | 25.62 | 58.03 | 41.48 | 23.15 | 22.91 | 23.73 | 34.8 | 37.0 |
| Llama 3-Instruct | 8B | 33.99 | 75.21 | 53.98 | 41.49 | 44.32 | 49.42 | 34.0 | 38.3 |
| Aya-23 | 8B | 51.11 | 73.65 | 54.37 | 36.39 | 43.76 | 42.28 | 31.6 | 32.0 |
| **ALLaM-Instruct** | 13B | 54.77 | 78.59 | 68.11 | 51.03 | 54.93 | 65.59 | 37.5 | 46.8 |
| SILMA-Instruct-v1.0 | 9B | 38.2 | 64.4 | 60.5 | 31.2 | 43.4 | 36.7 | 29.8 | 42.2 |
| Yi-1.5-Chat | 9B | 29.8 | 67.57 | 45.5 | 36.02 | 31.47 | 43.6 | 28.7 | 47.8 |
| AceGPT-Chat-v1.5 | 13B | 49.41 | 64.93 | 60.7 | 37.92 | 40.04 | 42.81 | 36.4 | 22.5 |
| Llama 2-Chat | 13B | 25.75 | 60.14 | 35.84 | 28.73 | 22.91 | 30.44 | 31.4 | 22.3 |
| Jais-Chat | 13B | 77.12 | 70.68 | 54.8 | 41.43 | 46.93 | 48.68 | 31.6 | 25.3 |
| **ALLaM-Instruct** | 70B | 57.91 | 79.01 | **75.92** | 62.23 | 58.47 | 78.38 | 38.4 | 56.8 |
| Jais-Chat-v1 | 30B | 80.52 | 71.14 | 60.4 | 43.99 | 48.6 | 48.52 | 32.9 | 26.0 |
| Jais-Chat-v3 | 30B | **88.37** | 70.05 | 62.37 | 30.15 | 51.21 | 38.53 | 37.3 | 32.5 |
| Qwen1.5-Chat | 32B | 37.39 | 78.86 | 57.25 | 50.62 | 48.23 | 59.73 | 39.0 | 43.0 |
| Yi-1.5-Chat | 34B | 30.85 | 65.96 | 45.6 | 35.47 | 35.2 | 40.22 | 25.3 | 49.8 |
| CommandR | 35B | 55.42 | 78.34 | 60.19 | 48.38 | 50.65 | 55.44 | 33.8 | 47.2 |
| Aya-23 | 35B | 55.56 | 79.69 | 57.71 | 47.78 | 51.77 | 56.18 | 33.8 | 43.8 |
| Mixtral-8x7B-Instruct-v0.1 | 47B | 37.91 | 77.27 | 52.66 | 41.09 | 42.64 | 49.37 | 32.5 | 39.7 |
| Llama 2-Chat | 70B | 30.72 | 59.49 | 40.77 | 32.86 | 28.68 | 30.6 | 32.3 | 25.5 |
| Llama 3-Instruct | 70B | 45.75 | 80.26 | 36.27 | 60.11 | 58.47 | 71.41 | 37.7 | 59.7 |
| Qwen1.5-Chat | 72B | 46.8 | **80.49** | 64.99 | 54.32 | 53.26 | 62.32 | 42.3 | 45.7 |
| Qwen2-Instruct | 72B | 51.9 | 79.98 | 71.51 | **66.18** | 58.66 | 75.16 | **47.7** | **61.7** |
| CommandR+ | 104B | 59.35 | 80.37 | 66.33 | 52.98 | 52.89 | 62.1 | 37.0 | 50.2 |
| DBRX-instruct | 132B | 45.75 | 76.46 | 56.6 | 46.73 | 48.79 | 53.17 | 30.5 | 48.8 |
| Mixtral-8x22B-Instruct-v0.1 | 141B | 43.79 | 76.45 | 58.92 | 46.74 | 49.72 | 55.55 | 35.1 | 46.0 |

Table 12: Comprehensive English benchmark results.

| | | AGIEval | MMLU STEM | MMLU Humanities | MMLU Social | MMLU Other | MMLU Average | MMLU-U-Pro | Ethics | WinoGrande | TruthfulQA | PIQA | ARC Easy | ARC Challenge | BoolQ | HellaSwag | TriviaQA | NQ Open | Minerva MATH | GSM8K | BBH | MixEval Hard | MixEval Standard | HumanEval pass@01 | HumanEval pass@10 |
|---|---|---|---|---|---|---|---|---|---|---|---|---|---|---|---|---|---|---|---|---|---|---|---|---|---|
| | | 0-shot | 0-shot | 0-shot | 0-shot | 0-shot | 0-shot | CoT 5-shot | 0-shot | 0-shot | 0-shot | 0-shot | 0-shot | 0-shot | 0-shot | 0-shot | 0-shot | 0-shot | 4-shot | 5-shot | CoT 3-shot | 5/0-shot (base /ft) | 5/0-shot (base /ft) | pass@0.1 | pass@0.10 |
| **Pre-trained** | | | | | | | | | | | | | | | | | | | | | | | | | |
| ALLaM-Base (from scratch) | 7B | 25.46 | 38.41 | 40.04 | 46.73 | 48.02 | 42.91 | 20.31 | 59.9 | 68.43 | 35.26 | 80.58 | 67.42 | 43.52 | 75.87 | 76.26 | 27.63 | 15.68 | 5.88 | 16.15 | 39.39 | 26.9 | 48.5 | — | — |
| ALLaM-Base | 7B | 24.19 | 34.73 | 37.43 | 45.53 | 46.94 | 40.71 | 20.43 | 44.15 | 69.0 | 35.22 | 79.0 | 67.36 | 43.65 | 75.77 | 76.17 | 42.1 | 13.96 | 5.52 | 16.98 | 41.28 | 20.4 | 50.2 | — | — |
| AceGPT | 7B | 26.44 | 30.07 | 35.52 | 37.73 | 38.24 | 35.38 | — | 48.89 | 69.3 | 38.39 | 77.75 | 72.64 | 44.11 | 75.66 | 75.43 | 49.64 | 18.39 | 2.84 | 11.14 | 36.81 | — | — | 12.28 | 20.23 |
| Llama 2 | 7B | 25.32 | 34.32 | 38.83 | 46.12 | 47.06 | 41.23 | 20.56 | 44.25 | 69.06 | 38.96 | 79.11 | 74.58 | 46.25 | 75.74 | 75.97 | 52.51 | 14.99 | 3.4 | 13.87 | 39.63 | — | — | — | — |
| Mistral-v0.3 | 7B | 32.85 | 49.54 | 52.84 | 68.57 | 66.85 | 58.65 | 31.67 | 64.49 | 73.56 | 42.57 | 82.15 | 78.24 | 52.13 | 82.14 | 80.38 | 60.68 | 20.42 | 12.38 | 37.53 | 58.27 | — | — | 27.82 | 42.12 |
| OLMo-1.7 | 7B | 30.58 | 39.11 | 43.23 | 54.18 | 52.66 | 46.79 | 18.57 | 60.47 | 69.14 | 35.91 | 80.14 | 75.04 | 45.14 | 80.89 | 77.28 | 15.35 | 11.41 | 5.98 | 27.67 | 34.07 | — | — | 17.66 | 27.21 |
| OLMo | 7B | 22.79 | 28.29 | 27.04 | 28.73 | 28.23 | 27.95 | — | 48.25 | 66.38 | 35.85 | 79.43 | 68.77 | 40.36 | 72.48 | 75.65 | 29.69 | 11.61 | 1.9 | 5.23 | 29.66 | — | — | 13.05 | 19.26 |
| Qwen2 | 7B | 45.09 | 64.57 | 60.55 | 80.79 | 73.73 | 69.25 | 40.84 | 65.24 | 72.38 | 54.25 | 81.18 | 74.41 | 49.74 | 72.48 | 78.82 | | | | 77.94 | 58.41 | | | | |
| Gemma | 8B | 38.6 | 54.57 | 54.62 | 71.14 | 68.19 | 61.16 | 34.24 | 56.09 | 73.01 | 45.49 | 80.9 | 80.81 | 53.24 | 82.84 | 80.72 | 63.35 | 16.4 | | 0.15 | | | | 32.48 | 52.2 |
| Llama 3 | 8B | 33.87 | 52.93 | 54.62 | 72.9 | 71.13 | 61.9 | 35.54 | 56.58 | 73.48 | 44.03 | 80.9 | 77.78 | 52.99 | 81.13 | 79.14 | | | 15.86 | 50.72 | 62.91 | | 57.5 | 27.5 | 44.61 |
| ALLaM-Base | 13B | 29.14 | 41.42 | 45.78 | 57.07 | 56.49 | 49.64 | 25.78 | 62.63 | 73.32 | 36.36 | 80.79 | 75.67 | 45.22 | 80.13 | 76.26 | 35.73 | 19.36 | 8.02 | 29.49 | 47.89 | 26.2 | 57.8 | **39.06** | **56.53** |
| Yi-1.5 | 9B | 37.8 | 63.88 | 60.91 | 79.1 | 73.61 | 68.87 | 40.58 | 63.35 | 73.01 | 46.67 | 80.79 | 79.12 | 54.69 | 85.81 | 77.78 | 54.32 | 17.87 | 29.96 | 64.82 | 70.97 | | | 17.44 | 27.46 |
| AceGPT-v1.5 | 13B | 32.4 | 42.06 | 48.33 | 59.64 | 58.48 | 51.65 | 26.01 | 59.25 | 72.38 | 38.07 | 80.2 | 76.26 | 50.77 | 80.7 | 80.1 | 60.89 | 15.6 | 6.5 | 23.96 | 46.32 | | 74.2 | 17.67 | 28.97 |
| Llama 2 | 13B | 32.47 | 40.95 | 47.08 | 58.56 | 57.39 | 50.5 | 25.86 | 60.93 | 72.14 | 36.89 | 80.52 | 77.44 | 48.98 | 80.58 | 79.39 | 60.89 | 14.93 | 5.54 | 23.43 | 47.35 | | | | |
| Jais | 13B | 24.12 | 29.27 | 33.39 | 34.32 | 35.79 | 33.2 | — | 61.77 | 68.51 | 35.39 | 77.91 | 70.08 | 41.81 | 67.74 | 71.76 | 33.45 | 9.61 | 4.18 | 11.45 | 36.86 | | | | |
| ALLaM-Base | 70B | 42.99 | 56.26 | 60.36 | 77.07 | 72.74 | 66.22 | 39.84 | 57.79 | 77.43 | 43.99 | 83.19 | 78.83 | 55.46 | 83.3 | 85.22 | 51.91 | 26.15 | 21.04 | 63.31 | 70.76 | **46.2** | **77.9** | 22.23 | 39.72 |
| Jais-v1 | 30B | 25.81 | 36.22 | 39.94 | 46.34 | 44.29 | 41.47 | — | 64.45 | 68.35 | 36.49 | 79.54 | 73.41 | 45.9 | 74.31 | 75.11 | 42.57 | 12.08 | 5.88 | 16.91 | 42.5 | | | | |
| Jais | 30B | 28.79 | 42.78 | 49.82 | 57.1 | 58.45 | 51.74 | — | 61.52 | 70.64 | 40.33 | 80.52 | 75.59 | 48.55 | 80.89 | 79.09 | 50.67 | 16.29 | 7.86 | 25.47 | 52.05 | | | 0.8 | 3.0 |
| Qwen1.5 | 32B | 47.02 | 66.67 | 64.06 | 81.87 | 77.95 | 71.62 | 44.7 | 67.37 | 74.27 | 57.46 | 82.75 | 71.38 | 51.02 | 87.31 | 83.7 | 66.18 | 24.35 | 37.56 | 74.91 | 53.6 | | | 28.09 | 53.72 |
| Yi-1.5 | 34B | 40.3 | 69.71 | 68.25 | 83.59 | 79.56 | 74.44 | 49.57 | 71.63 | 79.08 | 53.88 | 82.54 | 73.79 | 56.74 | 82.98 | 83.7 | 71.39 | 29.75 | 35.32 | 77.03 | 76.16 | 36.2 | 81.7 | 37.79 | 55.15 |
| Mixtral-8x7B-v0.1 | 47B | 39.62 | 58.99 | 60.62 | 78.26 | 74.22 | 67.13 | 41.76 | 63.21 | 75.61 | 48.5 | 83.57 | 83.38 | 59.81 | 85.26 | 83.88 | 71.39 | 29.75 | 27.18 | 58.53 | 69.36 | 42.5 | 77.0 | 30.28 | 46.67 |
| Llama 2 | 70B | 44.0 | 54.65 | 60.13 | 76.89 | 72.1 | 65.22 | 38.48 | 65.99 | 77.98 | 41.82 | 82.75 | 82.54 | 57.25 | 83.67 | 83.81 | 72.21 | 25.43 | 13.96 | 53.75 | 67.04 | 39.1 | 76.4 | 4.63 | 11.74 |
| Qwen1.5 | 72B | 48.04 | 68.71 | 67.95 | 86.68 | 81.66 | 75.46 | 53.88 | 72.61 | 80.11 | 59.63 | 82.64 | 84.6 | 61.25 | 85.26 | 84.95 | 77.84 | 23.66 | 37.94 | 81.12 | 82.12 | 45.6 | 74.6 | | |
| Qwen2 | 72B | 46.67 | 69.11 | 66.61 | 86.58 | 86.13 | 74.41 | 57.55 | 72.47 | 77.51 | 60.15 | 84.69 | 74.79 | 53.67 | 89.45 | 84.93 | | | 36.9 | 79.3 | 47.24 | 55.9 | 84.0 | | |
| DBRX | 132B | 57.05 | 79.58 | 77.62 | 89.7 | 86.13 | 82.59 | 57.55 | 76.97 | 79.16 | 54.73 | 85.42 | 86.32 | 67.06 | 89.45 | 88.3 | 80.85 | 40.19 | 49.08 | 88.78 | 79.67 | 33.8 | | | |
| Mixtral-8x22B-v0.1 | 141B | 44.39 | 63.02 | 63.06 | 80.99 | 78.08 | 70.3 | 44.65 | 65.69 | 76.64 | 55.63 | 85.57 | 84.18 | 60.15 | 90.15 | 85.57 | 77.32 | 29.7 | 30.46 | 71.42 | 67.52 | 28.9 | | | |
| | 14.1B | 46.84 | 66.67 | 63.98 | 83.98 | 78.88 | 73.92 | 51.11 | 66.71 | 79.87 | 52.2 | 84.93 | 84.18 | 63.14 | 87.89 | 86.25 | | 29.7 | 38.88 | 75.97 | 50.72 | | 81.5 | | |
| **Fine-tuned** | | | | | | | | | | | | | | | | | | | | | | | | | |
| ALLaM-Instruct (from scratch) | 7B | 46.35 | 54.27 | 55.77 | 71.95 | 69.36 | 61.99 | 30.34 | 69.35 | 69.93 | 41.31 | 80.3 | 75.0 | 52.05 | 82.94 | 75.4 | | | 13.02 | 53.6 | | 33.8 | 70.7 | 22.24 | 29.85 |
| ALLaM-Instruct | 7B | 47.09 | 49.95 | 52.88 | 67.21 | 66.21 | 58.31 | 27.78 | 69.8 | 70.56 | 42.11 | 79.82 | 76.22 | 51.45 | 78.93 | 75.2 | | | 12.84 | 49.28 | 41.28 | 28.9 | 67.6 | 17.65 | 27.08 |
| AceGPT-Chat | 7B | 26.33 | 38.34 | 42.47 | 49.89 | 53.81 | 44.15 | — | 53.38 | 66.38 | 49.34 | 75.24 | 68.01 | 42.32 | 73.84 | 70.92 | | | 3.0 | 13.04 | 35.99 | 30.8 | 61.7 | | |
| Llama 2 Chat | 7B | 35.55 | 36.41 | 43.19 | 53.01 | 54.84 | 46.4 | 22.87 | 58.88 | 66.38 | 45.32 | 77.26 | 69.74 | 44.28 | 79.76 | 75.52 | | | 4.86 | 23.35 | 40.15 | 36.2 | 70.0 | 36.59 | 52.55 |
| Mistral-Instruct-v0.3 | 7B | 42.22 | 50.68 | 54.56 | 69.52 | 67.14 | 59.75 | 36.33 | 73.8 | 73.8 | 59.65 | 82.59 | 82.62 | 58.7 | 85.81 | 82.88 | | | 13.28 | 48.67 | 56.78 | 26.7 | 55.0 | 16.6 | |
| OLMo-Instruct | 7B | 35.2 | 38.34 | 42.7 | 55.96 | 57.64 | 47.2 | — | 63.54 | 66.77 | 55.55 | 76.77 | 64.1 | 43.43 | 64.1 | 80.6 | | | 1.5 | 11.52 | 33.31 | | | | |
| Qwen2-Instruct | 7B | 53.72 | 63.91 | 62.93 | 80.37 | 77.74 | 69.78 | 46.79 | 71.62 | 69.69 | 57.31 | 80.63 | 76.64 | 54.1 | 85.41 | 80.6 | | | 27.02 | 77.86 | 59.39 | 39.1 | 69.6 | | |
| Gemma-it | 7B | 39.27 | 42.69 | 44.78 | 57.96 | 57.74 | 50.15 | 25.93 | 68.7 | 69.22 | 47.49 | 77.97 | 72.64 | 48.89 | 81.35 | 73.04 | | | 7.38 | 2.27 | | 45.6 | 75.0 | 11.52 | 71.92 |
| Llama 3-Instruct | 8B | 44.35 | 54.71 | 57.96 | 74.13 | 71.71 | 63.62 | 41.32 | 68.07 | 71.67 | 51.72 | 78.67 | 79.76 | 56.83 | 83.09 | 73.98 | | | 7.38 | 73.44 | 67.67 | | | 11.12 | 18.21 |
| Aya-23 | 8B | 35.1 | 37.8 | 47.91 | 57.55 | 57.23 | 47.97 | 24.79 | 58.79 | 63.06 | 45.28 | 81.4 | 74.11 | 55.89 | 84.77 | 73.18 | | | 6.46 | 42.15 | 48.07 | | | 18.22 | 19.16 |
| ALLaM-Instruct | 13B | 48.42 | 53.05 | 53.01 | 74.1 | 68.81 | 61.8 | 34.05 | 76.47 | 74.11 | 57.69 | 81.34 | 77.36 | 55.89 | 84.77 | 76.28 | | | 23.22 | 71.34 | 65.11 | 37.2 | 72.8 | 31.71 | |
| SILMA-Instruct-v1.0 | 9B | 38.71 | 46.31 | 48.46 | 54.27 | 53.81 | 50.43 | — | 58.36 | 50.75 | 56.87 | 80.09 | 64.9 | 40.78 | 83.24 | 63.08 | | | 4.54 | 65.66 | 54.69 | 40.9 | 74.2 | 55.1 | 67.62 |
| Jais-Chat | 13B | 55.93 | 66.51 | 62.66 | 81.9 | 72.9 | 69.46 | 47.29 | 71.46 | 74.43 | 52.49 | 77.26 | 55.46 | 42.32 | 86.61 | 79.22 | | | 38.28 | 78.85 | 68.61 | 51.2 | 81.0 | 21.77 | 35.22 |
| Yi-1.5-Chat | 34B | 38.92 | 69.58 | 70.1 | 83.49 | 79.95 | 75.1 | 31.3 | 71.98 | 73.16 | 62.16 | 79.32 | 80.64 | 52.13 | 82.87 | 80.51 | | | 7.2 | 81.5 | 75.07 | 45.2 | 81.7 | 65.39 | 79.64 |
| AceGPT-v1.5-Chat | 13B | 37.73 | 43.07 | 49.78 | 61.94 | 60.44 | 53.3 | 27.19 | 70.52 | 70.96 | 43.95 | 79.11 | 73.7 | 50.17 | 81.65 | 79.66 | | | 5.48 | 59.29 | 62.54 | | 77.0 | 15.54 | 20.02 |
| Jais-Chat | 13B | 31.45 | 38.42 | 47.48 | 55.28 | 56.87 | 49.46 | — | 64.92 | 68.51 | 39.66 | 78.24 | 69.4 | 46.84 | 83.81 | 77.6 | | | 6.46 | 24.64 | 38.1 | | | | |
| **ALLaM-Instruct** | 70B | 55.67 | 63.88 | 71.2 | 83.97 | 80.32 | 75.43 | 48.61 | 77.28 | 79.84 | 58.79 | 83.09 | 79.84 | 59.56 | 87.28 | 83.22 | | | 30.32 | 88.58 | 70.42 | 51.6 | 83.5 | 40.29 | 58.18 |
| Jais-Chat-v1 | 30B | 33.1 | 43.32 | 71.2 | 60.16 | 60.22 | 53.18 | — | 69.07 | 69.69 | 42.8 | 80.09 | 79.34 | 52.65 | 87.8 | 79.89 | | | 9.12 | 32.68 | 44.71 | | | | |
| Jais-Chat-v3 | 30B | 36.78 | 47.99 | 53.37 | 66.4 | 64.92 | 57.57 | 26.45 | 68.03 | 70.09 | 42.34 | 73.18 | 76.3 | 51.02 | 90.0 | 78.91 | | | 13.68 | 51.25 | 50.79 | 43.3 | 81.0 | 55.1 | 67.62 |
| Qwen1.5-Chat | 32B | 48.77 | 66.67 | 69.56 | 81.9 | 77.92 | 73.47 | 47.5 | 67.82 | 73.16 | 66.96 | 77.09 | 66.71 | 50.68 | 88.72 | 84.71 | | | 27.64 | 72.86 | 54.69 | 51.2 | 81.7 | 21.77 | 35.22 |
| Yi-1.5-Chat | 34B | 61.99 | 69.56 | 70.1 | 83.49 | 79.95 | 75.1 | 53.35 | 69.3 | 79.32 | 62.16 | 80.63 | 80.64 | 61.77 | 90.18 | 84.04 | | | 41.08 | 81.5 | 75.07 | 45.2 | 81.7 | 65.39 | 79.64 |
| Qwen1.5-Chat | 35B | 46.77 | 52.78 | 58.68 | 73.74 | 70.68 | 63.91 | 34.87 | 69.3 | 70.24 | 56.35 | 81.23 | 73.7 | 57.25 | 86.45 | 82.34 | | | 11.36 | 59.29 | 63.23 | | 77.0 | 15.54 | 20.02 |
| Aya-23 | 35B | 38.99 | 51.19 | 57.6 | 71.6 | 73.72 | 61.91 | 34.87 | 61.74 | 77.96 | 61.8 | 83.21 | 83.21 | 59.97 | 87.03 | 82.31 | | | 13.32 | 64.9 | 63.23 | 42.5 | 76.4 | 15.04 | 33.66 |
| Mixtral-8x7B-Instruct-v0.1 | 47B | 45.67 | 63.88 | 60.17 | 72.82 | 72.51 | 67.58 | — | 84.82 | 74.9 | 61.8 | 84.82 | 84.82 | 55.97 | 85.55 | 83.5 | | | 26.74 | 64.06 | 68.12 | 64.6 | 76.4 | 50.06 | 65.19 |
| Llama 3-Instruct | 70B | 63.78 | 72.53 | 72.82 | 85.29 | 83.52 | 78.58 | 59.52 | 77.03 | 75.53 | 52.77 | 86.96 | 84.81 | 64.33 | 87.49 | 82.49 | | | 44.12 | 91.05 | 88.15 | 55.9 | 84.0 | 50.06 | 29.59 |
| Qwen1.5-Chat | 72B | 52.17 | 70.47 | 72.99 | 84.66 | 81.46 | 76.86 | 49.52 | 78.24 | 77.11 | 63.86 | 78.84 | 82.2 | 61.52 | 88.14 | 86.19 | | | 21.88 | 81.12 | 52.19 | 48.3 | 84.1 | 38.93 | |
| Qwen2-Instruct | 72B | **66.00** | **79.42** | **77.94** | **88.89** | **86.13** | **82.29** | **62.48** | 75.54 | 80.75 | 59.64 | 84.0 | 82.2 | 61.52 | 89.2 | 86.65 | | | 41.2 | **89.54** | 46.51 | | 81.5 | | |
| CommandR+ | 104B | 45.97 | 60.23 | 65.33 | 80.31 | 75.41 | 69.7 | 45.22 | 61.78 | 65.82 | 59.64 | 71.27 | 61.78 | 46.33 | 85.81 | 86.65 | | | 25.34 | 76.04 | 52.19 | 51.4 | | | |
| DBRX-Instruct | 132B | 49.09 | 62.99 | 67.93 | 82.26 | 78.89 | 72.58 | 45.22 | 70.9 | 78.37 | **67.23** | **85.09** | **87.92** | 61.78 | 90.24 | 87.88 | | | 29.88 | 74.15 | 72.66 | | | 15.04 | 38.93 |
| Mixtral-8x22B-Instruct-v0.1 | 141B | 55.5 | 68.25 | 70.75 | 84.4 | 79.88 | 75.2 | 55.92 | **80.7** | **80.66** | **69.1** | 85.26 | 85.86 | **68.17** | 89.88 | **87.7** | | | 41.8 | 85.6 | 82.86 | | | | |

