# OpenReview forum: "ALLaM: Large Language Models for Arabic and English"
_ICLR.cc/2025/Conference — ICLR 2025 Poster_

### Official Review · Reviewer_SGGa · 2024-10-25

**Soundness:** 4
**Presentation:** 4
**Contribution:** 3
**Rating:** 8
**Confidence:** 3

**Summary:**

This paper describes the process for building ALLAM, an LLM trained on Arabic and English. Details about the training data and each stage of the training process are presented. Various evaluations are performed, including both automatic benchmarks like Arabic MMLU and human evaluation.

**Strengths:**

This is a timely paper, as many research groups are working on building LLMs for languages other than English. The paper presents useful details on their experience and will serve as a good reference.

**Weaknesses:**

There are no major weaknesses in my opinion. I would recommend adding discussing about other efforts in building LLMs for non-English languages and explaining how yours compare, i.e. moving some of the Related Work into the main paper. Especially it would be useful for the reader (who isn't referring to the appendix) to know about Jais and any other Arabic-centric model.

**Questions:**

1. Page 3 says 4T English tokens then and 5.2T tokens for 30B pre-training. It was a bit confusing what is the 5.2T. Also Table 1 lists English as 660B tokens from mixed corpora -- is this a subset from the 4T English Only column?

2.  Fig 5/6: for my display, there are black lines in the legend (e.g. Arabic/English MMLU) but only red/orange lines in the figure.

3. I am curious if you have considered using something other than Llama2 as your base model. If so, why did you pick Llama2 in the end?

---

> ### Author Response · Authors · 2024-11-14
>
> We thank the reviewer for their time and are grateful for their positive feedback, and we believe their comments helped us improve the paper’s clarity. We modified the paper to improve the its readability, with specific changes mentioned below.
>
> # Weaknesses
>
> **Mention of Other Models in the Main Manuscript**: The decision to move the detailed “related works” section to the appendix was due to page limitations. However, we appreciate reviewers' suggestions to mention some critical prior work (i.e., Jais, AceGPT) on the main paper. Therefore, we added the penultimate introduction section to address this valid concern.
>
> # Questions
>
> We appreciate the reviewer’s queries, and below is a response to each question.
>
> **Data mixture**
>
> > Page 3 says 4T English tokens then and 5.2T tokens for 30B pre-training. It was a bit confusing what is the 5.2T.
>
> The initial 4 trillion tokens were constructed purely from open-source repositories such as Dolma, Pile, Pes2o, and RefinedWeb. Before we started training our 30-billion-parameter model, RedPajama v2 (RPv2) released 84 snapshots of CommonCrawl data, along with rich metadata accompanying the text. With the new improved collection of the RPv2 data, we applied various metadata-based filtering criteria to extract 5.2 trillion tokens from the original 30 trillion token dataset. The data filtration criteria used to reduce the dataset from 30 trillion tokens to 5.2 trillion tokens are detailed in Tables 8 and 9. We have also explained more on “Why did we change the training data distribution for 30B experiments” in the FAQ section in the appendix. Please feel free to let us know if you have any further queries.
>
> > Also Table 1 lists English as 660B tokens from mixed corpora -- is this a subset from the 4T English Only column?
>
> Yes, this is a subset of the 4T tokens. Thanks for pointing this out. We have updated Table 1’s caption to clarify this.
>
> **Figure Legend**
>
> The legends in Figures 5 and 6 show what line colors and line styles mean. Line styles are shown with black lines. For example, in Figure 5, the red dashed line denotes the MMLU English score using randomly initialized embeddings, since the red color denotes randomly initialized embeddings, and the dashed line denotes English MMLU scores. If needed, we can add further clarification in the Figures’ caption.
>
> **Why Llama2?**
>
> When we began our initial training, the only Chinchilla optimal (with extended training) model available was LLaMA-2. In addition to that, LLaMA-2 models were not only open weight, it also came up with a paper explaining the complete data distribution of the training datasets which helped us doing some initial testing of the model and selection of on our own data mixtures. This is why we used the LLaMA-2 family of models. After training ALLaM models on top of LLaMA-2, while we started pre-training from scratch, we observed the emergence of several impressive open weight models, such as Mistral, Gemma, Yi, Qwen-2 etc.

---

### Official Review · Reviewer_9J4g · 2024-11-04

**Soundness:** 4
**Presentation:** 4
**Contribution:** 3
**Rating:** 8
**Confidence:** 5

**Summary:**

This paper presents a ALLaM: a family of large language models for Arabic and English, based on LLama-2 family of models.

The paper shows a method that can customize an existing LLM (like Llama) to acquire a much better performance at another language that was not included at all the original training of the model, without destroying the LLM's performance on previous language (i.e. second language acquisition).

The paper shows that second language acquisition can be done by augmenting the model in two major steps: 1-vocabulary expansion, and 2- carefully pre-training on mixture of new language and old language. This way the second language acquisition is not accompanied by catastrophic forgetting of the original language.

The paper includes a lot of experiments and comparisons with different baselines, including training from scratch. Showing that their second language acquisition method beats training from scratch. Many evaluations, both automated and human-evals are included in the paper.

**Strengths:**

Originality: As far as I know, this is not the first work to fine-tune llama-based models for Arabic or other language acquisition. In my opinion, the main originality source in this paper might be the data curation effort, and the vocabulary expansion and continual pre-training without loss of original language. For example, the finding that initializing the embedding of the new tokens from averages of smaller tokens in the original tokenizer is an very useful finding and I expect to see it being used more in the future.

Quality: The effort spent in human evaluation and curation of the data is notable, and I hope this dataset gets released, at least partially for the research community to build on. The comparison against other Arabic-enabled LLMs are also very comprehensive and useful.

Clarity: While many typos and non-standard terminology exist and are highlighted in the weaknesses section. The paper is still mostly easy to follow and read. The description of the datasets and training/evaluation procedure is particularly clear and concise.

Significance: Arabic is definitely an excellent language choice for this exploration of second language acquisition, given that the original LLama2 family of models do not train on Arabic at all (as shown in Table 10 in the original Lama2 report), and that means that the tokenizer and model are not very well trained for Arabic. Also, Arabic is definitely an underserved language in the LLM, especially given the size of its native speakers population (fifth most in the world, roughly 400M people).

**Weaknesses:**

Even though the paper is mostly well-written, it's littered with typos, non-standard terminology and missing references:
- "Fertility rate": very confusing term, it's not really a rate, and is usually referred to "fertility score": https://arxiv.org/pdf/2310.08754
- Figure 8 is not referenced anywhere as far as I can see.
- line 475: missing reference
-  line 294: missing reference
- line 291: lower case "we"
- line 072: training of "these"
- lines 151-153: DA identification is a hard task ... classifying the data to DA is not very difficult (opposite claim)
- line 155: in-house in-house

- I would have loved to see some newer Arabic models like Silma included in the comparisons for this paper, or a comparison against the models in the huggingface Arabic leaderboard.

**Questions:**

- How many Arabic tokens already existed in the LLama2 backbone, and how many were added after the expansion?

- Why did the training mix change for the 34B model specifically? (lines 286-292)

- I would be interested to compare allam against Silma Arabic llms

- Please consider evaluating on more dataset like AlGhafa benchmark or including ALLAM on the huggingface Arabic leaderboard.

---

> ### Author Response · Authors · 2024-11-14
> **Official Comment by Authors #1**
>
> We would like to express our gratitude to the reviewer for providing an in-depth response covering the big-picture contributions, minor typographical errors, and everything in between. We believe that the reviewer's comments, suggestions, and guidance have greatly helped improving the paper. We have addressed the reviewer's specific comments below.
>
> # Weaknesses
>
> Thank you for highlighting these typos and mistakes! We have addressed all of these issues in our paper.
>
> # Questions
> > How many Arabic tokens already existed in the LLama2 backbone, and how many were added after the expansion?
>
> We expanded Llama 2’s vocabulary from 32,000 to 61,586 tokens with all the added tokens being Arabic. We ran the following regex matching script to check the number of tokens that contain at least a single Arabic character:
>
> ```python
> from transformers import LlamaTokenizer
> import regex as re
>
> arabic_regex = r'^(?=.*\p{Arabic}).*$'
>
> def count_arabic_tokens(tokenizer: LlamaTokenizer) -> int:
>     return sum(re.match(arabic_regex, token) is not None for token in tokenizer.get_vocab())
> ```
>
> Llama 2 contained 46 Arabic Tokens (mostly single Arabic characters), while ALLaM contained 29,552 Arabic Tokens post-expansion.
>
> Note that Llama 2’s tokenizer does not include all Arabic letters, and thus has to rely on byte fallback for less common Arabic letters. For instance, it tokenizes the letter “ؤ” as two bytes `['<0xD8>', '<0xA4>']`.
>
> > Why did the training mix change for the 34B model specifically?
>
> Not all the training was done at the same time. As the training progressed, we gained more knowledge about our process, data, and the entire ecosystem of our training engine. Iterating over a single training run incurred significant costs, so we always prioritized quality over ablations for large-scale training runs. Given the available compute and deadline, we were able to conduct only one training run of the 34B model. We discovered that we could apply custom filters to a large data collection based on our use cases and preferences. In the first phase, we used an open-source data collection, and in the second phase, we filtered a collection of 84 Common Crawl snapshots down from approximately 30T tokens to 5.2T tokens. After filtering, we performed a few manual checks to verify the data quality. More details are included in the *Frequently Asked Questions* section in the appendix.

---

> > ### Author Response · Authors · 2024-11-14
> > **Official Comment by Authors #2**
> >
> > > I would be interested to compare allam against Silma
> >
> > It is promising to see other projects focusing on second language acquisition for LLMs, such as Silma adding Arabic to Gemma 9B. Our hope is that the findings for these projects can be combined through transparent research to further the advancements in Arabic and low-resource language LLMs.
> >
> > We have compiled the results of `silma-ai/SILMA-9B-Instruct-v1.0` through the same evaluation pipeline used in the paper. Here is the summary of the results:
> >
> > |Evaluation|silma-ai/SILMA-9B-Instruct-v1.0| ALLaM-7B (from scratch) |
> > |-|-| - |
> > |ACVA (5-shot)| 64.4 | 79.59 |
> > |araMath (5-shot)| 42.2 | 42.2 |
> > |araSwag (10-shot)| 38.2 | 50.98 |
> > |araTruthfulQA (0-shot)| 29.8 | 30.7 |
> > |ETEC (0-shot)| 36.7| 67.34 |
> > |Exams AR (5-shot)| 43.4| 52.89 |
> > |MMLU AR (Koto et al, 0-shot)| 60.5 | 69.16|
> >
> > Note that evaluation results are sensitive to many factors, such as the use of templates, prompt formatting, or evaluation method (CoT, Cloze, MCQ, etc…) [1-3]. We have custom implementations for these evaluations on the [LM Evaluation Harness](https://github.com/EleutherAI/lm-evaluation-harness), which might differ from other implementations. For `MMLU AR (Koto et al, 0-shot)` we use the [official implmentation](https://github.com/mbzuai-nlp/ArabicMMLU.git).
> >
> > > Please consider evaluating on more dataset like AlGhafa benchmark or including ALLAM on the huggingface Arabic leaderboard.
> >
> > HuggingFace’s Open Arabic LLM Leaderboard only accepts models with public weights on their website. We plan on releasing our ALLaM-7B (from scratch) model on HuggingFace within the next few weeks, which allows us to evaluate and submit our models to the leaderboard.
> >
> > ---
> > [1] Zheng, Chujie, et al. "Large language models are not robust multiple choice selectors."
> >
> > [2] Alzahrani, Norah, et al. "When benchmarks are targets: Revealing the sensitivity of large language model leaderboards."
> >
> > [3] Pezeshkpour, Pouya, and Estevam Hruschka. "Large language models sensitivity to the order of options in multiple-choice questions."

---

> > > ### Comment · Reviewer_9J4g · 2024-11-24
> > >
> > > Thank you for the detailed response. I am happy to increase the presentation score after fixing all the typos and missing references.
> > >
> > > I would suggest you add the Silma comparison to the paper or appendix for completeness.

---

> > > > ### Author Response · Authors · 2024-11-26
> > > >
> > > > We appreciate your follow-up! We have modified the latest manuscript to reflect your suggestions. You can see the other modifications in our reply to the rebuttal modifications thread.

---

### Official Review · Reviewer_gnB6 · 2024-11-06

**Soundness:** 3
**Presentation:** 3
**Contribution:** 2
**Rating:** 3
**Confidence:** 5

**Summary:**

This paper presents a series of LLMs for Arabic adapted from Llama2 as well as training from scratch.  The paper gives technical details of the pre-training, alignment and evaluation of these models.

**Strengths:**

The paper gives the details of the training of Arabic LLMs, including training from scratch and contunue training from Llama2.  The result Arabic LLMs achieve SoTA on Arabic benchmarks.

**Weaknesses:**

This is mainly a technical report rather than a research paper.  I do not see novel ideas or methods proposed.

**Questions:**

none

---

> ### Author Response · Authors · 2024-11-14
>
> Thank you for your review. We respectfully disagree with the statement that the paper is closer to a technical report rather than a research paper. Technical reports are typically only descriptive, with many obfuscated details, which may or may not carry significant details on novelty with the necessary information for reproducibility. However, we believe our paper presents novel ideas that aid in general LLM second language acquisition, and potentially even second modality acquisition. Our novel contributions are mentioned in the last paragraph of the introduction section, and are also highlighted independently by reviewer 9J4g. These include vocabulary expansion, embedding initialization, methods to maintain original language performance, effect of translation data, and data mixture considerations. We also added a detailed description of our data selection in Tables 8 and 9. Additionally, as opposed to technical reports, we tried to be as transparent as possible, mentioning our ablation results for translation, embedding initialization, and language mixing ratios, as well as detailed data descriptions, cleaning methods, and training logs. We also explained a lot of our decision-making process and incentives in the Frequently Asked Questions Section and added many details in the appendix for curious readers.
>
> Regardless of our best effort, we understand we can miss any details that the reviewer cares about. Also, we appreciate reviewers' disagreement and constructive feedback. During the rebuttal period, we are here to reply to all the queries of our respected reviewer.  If you want to know any specific technical details, please feel free to ask and we will put our highest effort to provide you the best possible reply.

---

### Official Review · Reviewer_hSix · 2024-11-09

**Soundness:** 3
**Presentation:** 3
**Contribution:** 3
**Rating:** 6
**Confidence:** 4

**Summary:**

This paper introduces ALLAM, a large language model for Arabic and English. The authors demonstrate the effectiveness of their training and alignment strategy for achieving state-of-the-art performance on several Arabic benchmarks.

**Strengths:**

Focus on Arabic Language: The paper specifically addresses the need for high-quality language models for Arabic, a language with a significant number of speakers worldwide and relatively fewer resources compared to English.  This focus fills a gap in the current landscape of large language models.

Comprehensive Training and Alignment Strategy: The authors employ a thorough approach to training and aligning ALLAM. They experiment with different data mixtures, vocabulary expansion techniques, and use both supervised fine-tuning and preference training to achieve optimal performance.

State-of-the-art Performance: ALLAM achieves state-of-the-art results on various Arabic benchmarks, demonstrating the effectiveness of the proposed techniques.  The authors also show that their model maintains or enhances English performance compared to the base Llama-2 model.

Exploration of Second Language Acquisition: The paper investigates the use of second language acquisition techniques for LLMs, which is a promising area of research.  The authors show how pretraining on a mixture of Arabic and English text can lead to effective learning of Arabic without catastrophic forgetting in English.

Detailed Methodology: The paper provides a detailed description of the training methodology, data curation process, and evaluation setup.  This transparency allows for reproducibility and facilitates future research in the field.

Commitment to Openness: The authors express their intention to make the ALLAM models openly available to the community, which promotes collaboration and further development of Arabic language models.

**Weaknesses:**

The authors should clearly state the originality of their work. While the paper builds on existing techniques such as second language acquisition for LLMs, it is not clear what specific novel components are being introduced. Is it the particular training recipe, the alignment strategy, or the focus on Arabic? A clear statement of originality is essential for establishing the contribution of this work.

Error Analysis:
The error analysis presented is currently insufficient. The authors need to go beyond simply stating that some evaluations provide more signal than others. A deeper dive into the types of errors made by the model on different benchmarks is necessary. What are the common patterns in incorrect predictions? Are there specific linguistic constructs or knowledge domains where the model struggles? Addressing these questions will provide valuable insights into the limitations of ALLAM and potential avenues for future research.


Additional Areas for Improvement:
Motivation for changing training data distribution: The authors mention changing the training data distribution for the 30B model. The motivation behind this change is not entirely clear and should be elaborated upon.
Fair comparison of models: The paper touches upon the difficulty of comparing models due to variations in training data size, architecture, and other factors. While comparing to larger models is a reasonable approach, the authors could consider additional metrics or analyses to further ensure a fair comparison.
Instruct model vs. base model results: The authors justify reporting results for the instruct model instead of the base model. However, providing some comparison or insights into the performance difference between the base and instruct models could be beneficial.

**Questions:**

...

---

> ### Author Response · Authors · 2024-11-14
> **Official Comment by Authors #1**
>
> We thank the reviewer for taking the time to provide very detailed and constructive feedback. We believe that addressing all the reviewer’s concerns, both here and in the paper, will not only enhance the quality of the paper but also improve its reproducibility.
>
> # Weaknesses
>
> **Statement of Originality** We agree with your assessment that all research should have a clear statement of originality. Therefore, the last paragraph of the introduction now enumerates the contributions of our work. Additionally, reviewer 9J4g independently mentions sources of originality in their assessment, in which they mention data curation methods, vocabulary expansion, and embedding initialization as novel contributions.
>
> **Error Analysis** We acknowledge the need for a more granular error analysis and appreciate your suggestions on the direction it could take. Given the complexity of the Arabic language, identifying specific linguistic constructs or domains where the model exhibits challenges remains an active focus in our ongoing research. We aim to expand this analysis by categorizing errors, particularly in areas such as linguistic complexity and domain-specific knowledge within fields like politics, history, and STEM.
> While time constraints prevented a full exploration in this submission, we are actively working on developing a taxonomy of error types which will inform subsequent iterations of ALLaM. We look forward to incorporating these insights in coming days during rebuttal, providing a more comprehensive view of the model's strengths and areas for refinement.
>
> For human evaluation, we did perform a comprehensive evaluation of common failure cases, such as ethical dilemmas, middle eastern culture, religions, illegal activities, human rights, locale awareness, and personality. We described in the *data* paragraph of Section 3.2 how we targeted these issues through our preference tuning approach.
>
> # Additional Areas for Improvement
>
> **Motivation for Changing Training Data Distribution** We tried addressing this question in the *Frequently Asked Questions* section in the appendix. We are sorry that we couldn’t add this to the main section of the paper due to page limitations.
>
> Not all the training was done at the same time. As the training progressed, we gained more knowledge about our process, data, and the entire ecosystem of our training engine. Iterating over a single training run incurred significant costs, so we always prioritized quality over ablations for large-scale training runs. Given the available compute and deadline, we were able to conduct only one training run of the 34B model. We discovered that we could apply custom filters to a large data collection based on our use cases and preferences. In the first phase, we used an open-source data collection, and in the second phase, we filtered a collection of 84 Common Crawl snapshots down from approximately 30T tokens to 5.2T tokens. After filtering, we performed a few manual checks to verify the data quality.
>
> **Fair Comparison of Models** Thanks you for acknowleding the difficulty of the fair evaluation of the model. We do agree that comparing our models with larger models is not an ideal solution. We believe the ideal metric would be measuring model performance given a fixed model size and fixed Arabic dataset size. However, given the opacity of data mixture information in most top models as well as number of token used to train the model, it is impossible to measure such a metric. Even if we were able to obtain the number of Arabic tokens for other models, we would not be able to control for data quality. Therefore, given these constraints, we were only able to compare our models with larger models to compensate for a potential data advantage.
>
> In our work, we mainly used,
> (i) Automatic Evaluation
> (ii) LLM as a judge
> (iii) Human evaluation
>
> We have added a wide range of evaluation on the different benchmarks in the appendix (table 10, 11) . We are very serious about doing our absolute best for the evaluation and fair comparison of the models so that we can continue building the state-of-the-art models. We are constantly looking for new ways to evaluate the models. If the reviewer has any actionable suggention on adding new analysis and/or using new metric for fair comparison, we'd love to do it during the rebuttal. Please let us know.

---

> ### Author Response · Authors · 2024-11-14
> **Official Comment by Authors #2**
>
> **Instruct Model vs. Base Model Results** Similar to the data distribution suggestion, we had to move this discussion to the appendix’s *Frequently Asked Questions* section due to page limits.
>
> The reason we reported the Instruct model results instead of the base model in the main paper is twofold:
>
> 1. **Blurred distinction between base and instruct models**: Modern pretraining often involves incorporating supervised fine-tuning (SFT) data, including alignment data designed to improve user interactions. Consequently, the distinction between a base model and an instruct model has become increasingly unclear. Many models today are pre-trained with some degree of alignment, making it difficult to evaluate them purely as base models.
>
> Here is an insight from our evaluation results.
>
>    For instance, as shown in Table 11, the Qwen2-7B-base achieves a score of 77.94 on GSM8k, while ALLaM-7B-base (trained from scratch) achieves 16.98, with a significant delta of 60.96. After supervised fine-tuning, ALLaM-7B-instruct (from scratch) scores 53.6, while Qwen2-7B-instruct scores 77.86, reducing the delta to 24.26. Although ALLaM shows an improvement of 36.62 points during SFT, Qwen2 experiences no notable gain in performance in this phase. We suspect this is due to the inclusion of alignment data during Qwen2's pretraining phase. Thus, while Qwen2 models are inherently better than ALLaM, the performance gap between Qwen2 and ALLaM at the base level does not necessarily reflect true model capabilities due to the suspected presence of alignment data in the base model.
>
> We've added these new discussions in the *Frequently Asked Questions* section of the paper.
>
> 2. **Focus on user interaction**: The primary goal of building these models is to optimize them for user interaction. Since users will interact with the instruct version of the model, it makes sense to report the results of the model in its instruct phase. This ensures that the reported performance is reflective of the actual experience users will have, making the results more relevant and impactful for the paper’s audience.

---

> ### Author Response · Authors · 2024-11-24
> **Additional ALLaM Error Analysis**
>
> We thank the reviewer for their valuable comment. Following up the suggestion of further error analysis, we identified several patterns where the model's performance could be refined through extensive human evaluation. These observations can be summarized as follows:
>
> 1. **Repetition Issues**: In certain versions, the model exhibited a tendency to overgenerate or repeat content. This behavior varied with temperature settings. After extensive testing, we found that a temperature of 0.6 yielded the most balanced outputs.
> 2. **Translation Challenges**: The model sometimes regenerates the translation instruction within the translated output, or attempts to summarize the shared context while translating it, rather than focusing solely on the translation itself.
> 3. **Variation in Outputs**: When tasked with summarization or translation, the model provides multiple outputs.
> 4. **Precise Instruction Following**: The model does not always adhere to specified output length or structure. For instance, when requested to summarize a text in 100 words, it might exceed this limit or produce a summary longer than the input text.
> 5. **Complex Arabic Proofreading**: The model faces challenges with certain Arabic-language tasks, such as applying proper punctuation, grammar, and diacritics. While it may provide partially correct answers, it often fails to deliver fully accurate or comprehensive outputs in these areas.
>
> Below are examples of failure cases. For brevity, we omitted parts of the questions/answers with "[...]" to focus on the failure cases.
>
> | Question | ALLaM's Answer | Error |
> |----------|------------|--------|
> | ماهي اختصاصات الأمن الوطني السعودي؟ | اختصاصات الأمن الوطني السعودي تشمل حماية الدولة وصون حدودها [...] وتطوير وتحسين كفاءة البنية التحتية والأداء في مجالات الأمن الوطني والتحريات والمعلومات وتطوير وتحسين كفاءة البنية التحتية والأداء [repeated many times] | Excessive repetition of the same phrase "وتطوير وتحسين كفاءة البنية التحتية والأداء" throughout the response. We noticed that the KL divergence penalty scalar in DPO impacts this. This issue occurs in fewer than 1% of our total evaluation samples. |
> | Have you ever struggled with persuading someone? [...] First, you need to work on your confidence [...] Finally, build trust [...] هلا تساعدني في ترجمتها باللغة العربية | بالطبع! يسعدني مساعدتك في ترجمة المقالة لتتمكن من الاستفادة منها في تدريس طلابك. يرجى تزويدي بنص المقالة وسأعمل على ترجمتها إلى اللغة العربية بكل سرور. | The model asked for context that was already provided. |
> | من فضلك، سألقي درس عن آخر آية [...] وأريد ترجمة تفسيرها من تفسير السعدي: {الْيَوْمَ أَكْمَلْتُ لَكُمْ دِينَكُمْ} بتمام النصر [...] فقوموا به شكرا لربكم | في هذا التفسير للآية الكريمة [...] يقول الشيخ السعدي إن هذه الآية تشير إلى أن الدين الإسلامي قد تم تكميله [...] | Instead of translating the provided Tafseer text, the model created a new explanation/summary of it. |
> | .فضلا هل يمكنك تقديم خدمة لي بترجمة هذا النص إلى اللغة الإنجليزية؟ تعرف البلاغة على أنها فن الإقناع [...] في تاريخ البلاغة | Please, can you do me the favor of translating this text into English? Rhetoric is defined as the art of persuasion [...] in the history of rhetoric | The model translated the request for translation instruction along with the content text. |
> | Write a poem in arabic | بالطبع يمكنني مساعدتك في كتابة قصيدة باللغة العربية. إليك قصيدة قصيرة كمثال: في حبّ الوطن يا وطني [...] | ALLaM provided Arabic introduction text instead of starting with English response before the poem. |

---

> > ### Author Response · Authors · 2024-11-27
> >
> > We truly appreciate our reviewers' time and effort. If there is any way we can help clarify any confusion for our reviewers, we would be happy to assist. Please let us know if you have any feedback on our replies.

---

### Author Response · Authors · 2024-11-14
**Rebuttal Revision November 13 19:00 AoE**

We sincerely thank the reviewers for their time and comprehensive reviews. We believe the reviews are general positive, highlighting the benefits of our novel methods (vocabulary expansion, embedding initialization, etc...) for second language acquisition, as well as our detailed training details that could benefit the training of similar models in the future. We have clarified the reviewer's questions and updated the manuscript on November 13 19:00 AoE according to their suggestions, which includes:

1. Moving Figure 8 (now Figure 12) to the appendix, next to its reference.
2. Addressing typos and missed references mentioned by reviewer 9J4g.
3. Addressing other minor typos after further review.
4. Modifying appendix tables to have consistent style and remain within the manuscript’s margins.
5. Adding the penultimate introduction paragraph to mention similar Arabic-centric LLMs and reference the detailed descriptions in the appendix.
6. Modifying Table 1’s caption to clarify that the mixed English dataset is a subset of the English only dataset.
7. Adding new result analysis on "Why did we report Instruct model result instead of base model result in the main paper?" in the Appendix B

---

> ### Author Response · Authors · 2024-11-26
> **Rebuttal Revision November 25 23:00 AoE**
>
> Following the review comments and the ongoing discussion, we have added a few modifications to the manuscript in addition to the previously mentioned modifications. These include:
>
> 1. Adding questions in the *Frequently Asked Questions* section (Appendix B):
> 	- "What are failure cases of ALLaM?" Along with Table 7 showcasing examples.
> 	- "How many Arabic tokens were in Llama 2, and how many did you add to ALLaM?"
> 2. Modifying Tables 11 and 12 to include the SILMA model.
> 3. Fixing clerical errors for ALLaM 34B in Table 4.
> 4. Other minor organizational edits.

---

### Author Response · Authors · 2024-11-24
**Furthering Discussion**

Seeing as the discussion deadline is approaching, we would appreciate any comments or insights to further the discussion and improve the manuscript.

---

### Meta-Review · Area_Chair_m4KH · 2024-12-07

**Metareview:**

This paper presents the process used to train ALLaM, a series of Arabic and English LMs finetuned on top of Llama-2. The paper considers several adaption techniques to adapt the model to a new language and provides insights into best practices for adapting English LMs to other languages in the future. The resulting models achieve good performance on the considered downstream tasks.

Strengths:
- The paper is well-motivated (Arabic is a relatively underserved language among LLMs), and the paper clearly describes the process and methodology used to adapt an existing LM to a new language.
- Based on the experimental results, ALLaM models achieve state-of-the-art performance on multiple Arabic language benchmarks.

Weaknesses:
The main contribution of this paper is (and is stated to be) the development of a family of Arabic language models that achieve the state-of-the-art on Arabic language benchmarks.  However, limited artifacts from the paper have been released: the 7B and 13B models are only available through (separate) proprietary platforms (Appendix B), and there is no firm agreement to fully release the models upon publication. The data collected for Arabic pretraining, supervised finetuning (SFT), and preference optimization (DPO)—another major contribution, in my opinion—has also not been released. I encourage the authors to fully release their models and data, which will provide a significant contribution to the Arabic NLP community.

Some issues raised by the reviewers, such as presenting the novel aspects of the paper more clearly (Reviewer hSix) and the typos raised by 9J4g, have been addressed in the revised version of the paper.

**Additional Comments On Reviewer Discussion:**

The authors provided extensive responses to the reviews, as well as revisions to the paper. Some reviewers did not participate in the discussion period (hSix, gnB6, SGGa); 9J4g increased their presentation scores in response.

---

### Decision · Program_Chairs · 2025-01-22

Accept (Poster)